

# Measurements of PM$_{2.5}$ with PurpleAir under atmospheric conditions

Karin Ardon-Dryer[1], Yuval Dryer[1], Jake N. Williams[1] and Nastaran Moghimi[2]

[1]Department of Geosciences, Atmospheric Science Group, Texas Tech University, TX
[2]Thomas S. Wootton High School, North Potomac, MD

*Correspondence to*: Karin Ardon-Dryer (karin.ardon-dryer@ttu.edu)

**Abstract.** The PurpleAir PA-II unit is a low-cost sensor for monitoring changes in the concentrations of Particulate Matter (PM) of various sizes. There are currently more than 9000 PA-II units worldwide; some of them are located in areas where no other reference air monitoring system is present. Previous studies have examined the performance of these PA-II units (or the sensor within them) in comparison to a co-located reference air monitoring system. However, because PA-II units are installed

by PurpleAir customers, the PA-II units are not co-located with a reference air monitoring system and, in many cases, are not near one. This study aimed to examine how PA-II units perform under atmospheric conditions when exposed to a variety of pollutants and PM$_{2.5}$ concentrations. We were interested in knowing how accurate these PA-II units are when measuring PM$_{2.5}$ concentrations with their sensitivity to concentration changes in comparison to the Environmental Protection Agency (EPA) Air Quality Monitoring Stations (AQMS) that are not co-located with them. For this study, we selected eight different locations,

where each location contains multiple PA-II units (minimum of seven per location, a total of 86 units) and at least one AQMS (total of 14). PM$_{2.5}$ measurements from each PA-II unit were compared to those from the AQMS and other PA-II units in its area. The comparisons were made based on hourly and daily PM$_{2.5}$ measurements. In most cases, the AQMS and PA-II units were found to be in good agreement; they measured similar values and followed similar trends, that is, when the PM$_{2.5}$ values measured by the AQMS increased or decreased, so did those of the PA-II. In some high-pollution events, the PA-II measured

higher PM$_{2.5}$ values compared to those measured by the AQMS. We found PA-II PM$_{2.5}$ measurements to remain unaffected by changes in temperature or Relative Humidity (RH). Overall, the PA-II unit seems to be a promising tool for identifying relative changes in PM$_{2.5}$ concentration with the potential to complement sparsely distributed monitoring stations and to aid in assessing and minimizing the public exposure to PM, particularly in areas lacking the presence of an AQMS.

## 1. Introduction

Atmospheric particulate matter (PM) with an aerodynamic diameter smaller than 2.5 μm (PM$_{2.5}$) is one of the leading contributors to the global burden of disease (GBD, Cohen et al., 2017; Forouzanfar et al., 2015; Lim et al., 2012). These particles are small enough to penetrate deep into the human lungs (Ling and van Eeden, 2009), where they have a negative impact on human health (Shiraiwa et al., 2017). Exposure to high PM$_{2.5}$ concentrations was found to be correlated with the

daily number of hospitalizations and mortality cases (Schwartz et al., 1996; Klemm and Mason, 2000; Di et al., 2017). In the US, 3 %–5 % of annual deaths are attributed to PM$_{2.5}$ (Cohen et al., 2017). Determining the pollution-level PM$_{2.5}$ exposure can





be challenging as a limited number of in-situ instruments are available for monitoring ground-level PM$_{2.5}$ concentrations (Ford et al., 2019).

In the United States, the Environmental Protection Agency (EPA) monitors ambient PM$_{2.5}$ concentrations by using air quality monitoring stations (AQMSs). These stations use equipment that implements either a federal reference method or federal equivalent method (FRM and FEM, respectively; Clements et al., 2017). The FRM is a gravimetric measurement in which particles are collected on a filter and the difference in filter weight before and after exposure is used to determine the 24-h PM concentration (Watson et al., 2017). The FEM measures PM using optical, beta ray attenuation and trapped element oscillation

to provide hourly PM concentrations. A single FEM PM$_{2.5}$ sensor in each AQMS costs thousands of dollars. Further, the operation of these AQMSs requires trained personnel and significant infrastructure; they are subject to strict maintenance and calibration routines to ensure high-quality data and comparability between different locations (Castell et al., 2017). AQMSs generally have sparse geographic coverage and are located at fixed sites, mainly in large population centers; they are not present in smaller cities and underdeveloped regions. The high temporal and spatial resolution of PM$_{2.5}$ concentrations may

vary significantly within a region, therefore, PM$_{2.5}$ concentration values provided by a single AQMS site may not accurately represent the PM$_{2.5}$ concentrations present near people who are concerned about their possible health effects (Wang et al., 2015). These limitations create a growing need for air quality sensor networks that will produce both temporal and spatial high-resolution pollution maps that can be used to identify peak events across large areas (Morawska et al., 2018).

Recent advancements in technology and a rise in public awareness have led to an increase in the popularity of low-cost air-quality sensors that are relatively cheap and easy-to-use (Commodore et al., 2017; Woodall al., 2017). Such sensors enable communities and individuals alike to obtain granular information on the spatial and temporal distribution of PM concentrations in their area (Gupta et al., 2018; Morawska et al., 2018), thereby enabling them to monitor local air quality conditions (Williams et al., 2018). Many types of low-cost air-quality sensors are available, and they vary in performance (Williams et al., 2018);

however, despite the proposed benefits of these sensors, their accuracy and precision remain unknown (Kuula et al., 2017). Data quality remains a major concern that hinders the widespread adoption of low-cost sensor technology. To assure data quality, it is important to test these sensors and compare them to FRM/FEM measurements under both laboratory and field conditions, particularly under atmospheric conditions with various air pollution levels in which the sensors are expected to operate (Kelly et al., 2017; Morawska et al., 2018). Testing these sensors at multiple locations will allow for exposure to

different atmospheric conditions and pollutant types (AQ-SPEC, 2018).

Among the limitations of low-cost sensors are environmental factors that affect the sensor's abilities. Some low-cost sensors have exhibited sensitivity to temperature and relative humidity (RH) (Clements et al., 2017). When working in the laboratory, these environmental conditions can be controlled; however, it is impossible to achieve such stability in the field under

atmospheric conditions. Therefore, additional measurements under a variety of ambient conditions are needed (Kelly et al.,



2017). In addition, some sensors have exhibited a drift in sensitivity over time (reduction of efficiency). The rate of drift over time is a crucial parameter in sensor characterization as it determines the interval of calibration as well as the overall useable lifetime of the sensor (Clements et al., 2017; Hagan et al., 2018).

The PA-II unit is a low-cost sensor sold by PurpleAir company. It is meant for outdoor usage and is the subject of our study. Each PA-II unit contains two Plantower particulate matter sensors (PMS5003 sensors) that provide real-time measurements of $PM_{1.0}$, $PM_{2.5}$, and $PM_{10}$. The usage of PA-II has grown rapidly in the last two years with the result that more than 9000 such sensors are in use across five continents, with the majority being operated in the US and Europe. PurpleAir provides live information on their website in the form of a color-coded air quality index (AQI) together with actual PM concentrations
(PurpleAir, 2019). Several studies have already evaluated the PA-II unit or the sensors (PMS5003) it contains; however, in all such studies, the PA-II unit (or the PMS5003 sensor) was co-located with a reference unit. The AQ Sensor Performance Evaluation Center (AQ-SPEC) evaluated the performance of a PA-II unit using FEM sensors as reference under laboratory and field conditions in the Los Angeles area. Their evaluation showed a very good comparison between the two for both $PM_{2.5}$ and $PM_{10}$ (AQ-SPEC, 2018). An additional comparison between three different PA-II sensors and a single FEM was performed
for eight weeks between December 2016 and January 2017 at the South Coast Air Quality Management District Rubidoux Air Monitoring Station. Good correlation ($R^2 > 0.9$) was found between the three PA-II units and the FEM unit. However, although the PA-II unit follows diurnal and day-to-day fluctuations very well, it consistently overestimated the $PM_{2.5}$ concentrations measured by the FEM (Gupta et al., 2018). Sayahi et al. (2019) conducted a long-term comparison (320 days) between two PMS5003 sensors and both FRM and FEM units that were all co-located at Salt Lake City, Utah. One of their PMS5003
sensors overestimated the $PM_{2.5}$ concentration whereas the other measured similar values to those measured by the FEM. According to Gupta et al. (2018), the performance of PA-II compared against FEM units in a high-pollution environment ($PM_{2.5} > 100$ µg m$^{-3}$) is unknown and requires further evaluation. In addition, the sensitivity of the PA-II sensors to changes in RH, temperature, and other environmental parameters remains a topic of further investigation (Gupta et al. 2018). Answers to these questions are crucial if we are to assess the possibility of using measurement data from multiple PA-II units to properly
represent the air quality of an area, thus allowing the residents to protect themselves when high pollution events occur.

    This study aimed to examine how PA-II units perform under atmospheric conditions when exposed to a variety of pollutants and $PM_{2.5}$ concentrations. Comparison of PA-II units to $PM_{2.5}$ measurements taken by an AQMS that was not co-located with them are presented. Further, a comparison of PA-II units to other nearby PA-II units and their efficiency as a network of low-
95 cost sensors are discussed.

## 2. Method

### 2.1. PurpleAir PA-II Unit Structure and Data



The PurpleAir PA-II unit has size of 85 × 125 mm. It contains two PMS5003 sensors (see two blue rectangles in Fig. 1A), a

BME280 environmental sensor, and an ESP8266 microcontroller. The BME280 sensor is used to monitor the units' inner pressure, temperature, and humidity; the sensor measurements are not to be used for monitoring ambient conditions (PurpleAir, personal communication, 2019). The ESP8266 microcontroller is used to communicate with both the two PMS5003 sensors and with the PurpleAir server over Wi-Fi, thereby allowing the PM concentration to be presented live on the PurpleAir map (https://www.purpleair.com/map). The PMS5003 sensors provide real-time measurements of $PM_{1.0}$, $PM_{2.5}$, and $PM_{10}$

concentrations; the sensors are based on the light scattering principle, and a photodiode detector converts the scattered light to a voltage pulse. A fan draws the particles into the sensor and past the laser path (Fig. 1B) at a flow rate of 0.1 L/min. The particle count is calculated by counting the pulses from the scattering signal and converting the number of pulses to a mass concentration for six diameters between 0.3 and 10 μm using an algorithm for outdoor PM (CF_ATM - average particle density). Each PMS5003 sensor has an effective measurement range for $PM_{2.5}$ concentration of 0–500 μg m$^{-3}$ with a resolution

of 1 μg m$^{-3}$, and the maximum standard $PM_{2.5}$ concentration is above 1000 μg m$^{-3}$ According to the manufacturer, each PMS5003 sensor will work effectively in a temperature range of -10 °C to 60 °C and RH range of 0 %–99 % (Yong, 2016).

The microcontroller in the PA-II unit reads the $PM_{1.0}$, $PM_{2.5}$, and $PM_{10}$ concentrations from the PMS5003 sensors every second; it averages the concentration values across 20 s and displays the results using UTC time (PurpleAir, personal communication,

2019). The use of a dual PMS5003 sensor setup serves as an internal check for the PA-II unit's integrity. The similarity/difference in the PM concentrations obtained from the two PMS5003 sensors (named as A and B) allows users to evaluate the efficiency and validity of their PA-II unit. The two PMS5003 sensors, A and B, should agree with each other all the time; failure to report the same value indicates that something is wrong with one of the sensors. PurpleAir does not calibrate their devices; instead, before each PA-II unit is sent out to a customer, the company performs a comparison test with a dozen

other PA-II units to find and remove outliers from the shipment (PurpleAir, personal communication, 2019).

All the data regarding the PA-II units and their measurements was downloaded from the PurpleAir website. Information about all the PA-II units was downloaded in a JSON formatted file. Each PA-II unit has a name (given by the owner), a unique ID number (designated by the company for each sensor), the unit location (latitude and longitude), and a date on which the unit

was installed. We initially selected all the PA-II units that were active between January 1, 2017, and December 31, 2018 (UTC time). For each selected PA-II unit, we downloaded an Excel file containing the measurement data in 20-s intervals for both PMS5003 sensors (A and B). Because our focus was on $PM_{2.5}$ measurements, we calculated the $PM_{2.5}$ hourly average and standard deviation (SD) based on the original measurement values and the daily average and standard deviation based on hourly averages that we had calculated previously. Our final dataset included only days that had a minimum of 20 h of

measurements per day (80 % of the day). Only times which had a good agreement ($R^2 > 0.9$) of hourly $PM_{2.5}$ measurements between the two PMS5003 sensors (A and B) were used.



## 2.2. PM$_{2.5}$ Measurements from AQMS

Hourly measurements of PM$_{2.5}$ (FRM/FEM Mass code - 88101 file) from all AQMSs collected by the EPA from January 1,
2017, to December 31, 2018, were selected from the EPA website (https://aqs.epa.gov/api). The location of each AQMS was
provided in the same file. Each AQMS is identified by the combination of state code, county code, site number, and Parameter
Occurrence Code (POC) number. The POC is used to represent cases in which more than one unit performs PM$_{2.5}$
measurements at the same site. All timestamps were converted to UTC to match the PA-II measurement timestamps. The PM$_{2.5}$
daily average and standard deviation were calculated based on the hourly PM$_{2.5}$ measurements; only days with a minimum of
20 h of measurements per day (80 % of the day) were considered.

## 2.3. Identification of Locations for Analysis - Areas with Multiple PA-II units and at least one AQMS

By using the JSON file for the PA-II and the 88101 file for the AQMS, the distances between all units was calculated to
identify locations with multiple PA-II units (a minimum of five units) and at least one AQMS. All the units in these locations
needed to be active during the designated time period of January 1, 2017, to December 31, 2018. Eight different locations
containing a total of 14 different AQMSs and 86 different PA-II units were identified: Pittsburgh, PA; Denver, CO; Berkeley-
Oakland, CA; San Francisco, CA; Vallejo, CA; Ogden-South Ogden, UT; Lindon-Orem, UT; and Salt Lake City, UT. Fig. S1
shows a map with all the PA-II units and AQMSs at each location. Table 1 provides information on each of the eight locations
with the names of the units, their location, first and last time of measurement, and the minimum and maximum PM$_{2.5}$ hourly
values.

In Pittsburgh, two AQMSs (42-3-8-3 and 42-3-1376-1) and eleven PA-II units (ID - 3723, 3981, 9016, 9026, 9038, 9096,
9878, 9880, 9892, 9896, and 9906) were used. In Denver, three AQMS (8-31-26-3, 8-31-27-3, and 8-31-28-3) and eight PA-
II units (ID - 2249, 2267, 2269, 2719, 2900, 3924, 4022, and 7956) were used. In in Berkeley-Oakland, three AQMSs (6-1-
11-3, 6-1-12-3, and 6-1-13-3) and ten PA-II units (ID - 2574, 3082, 3854, 4335, 4506, 4795, 4825, 5414, 6410, and 10114)
were used. San Francisco, Vallejo, Ogden-South Ogden, and Lindon-Orem all had a single AQMS (6-75-5-3, 6-95-4-4, 49-
57-2-5, and 49-49-4001-5, respectively) but multiple PA-II units. San Francisco had nine PA-II units (ID - 1226, 2031, 2910,
3348, 3996, 4372, 4770, 5776, and 6344); Vallejo had 15 units (the maximum; ID - 1142, 1870, 1874, 1878, 1882, 2480, 2906,
3686, 3758, 3769, 3782, 3784, 3960, 4928, and 5127); Ogden-South Ogden had seven PA-II units (the minimum; ID - 465,
1104, 5178, 5454, 6604, 7858, and 7860); and Lindon-Orem had 12 PA-II units (ID - 5135, 5143, 5145, 5728, 5732, 5736,
5750, 5754, 5760, 6304, 6948, and 6986). Salt Lake City had two AQMSs at the same location (49-35-3006-4 and 49-35-
3006-5, different POCs) and 14 PA-II units (ID - 884, 3388, 5014, 5460, 5742, 5802, 5990, 6078, 6356, 6360, 6434, 6608,
6622, and 10050).

## 2.4. Comparison between PA-II and AQMS



To evaluate the similarities and differences between the AQMS and the PA-II units, a set of calculations and comparisons was performed. First, graphs showing the distribution of PM$_{2.5}$ values were plotted. Second, a regression between the AQMS and each PA-II unit was made based on hourly and daily PM$_{2.5}$ measurements. From the regression, R-squared (R$^2$) and root mean square error (RMSE) values as well as the best fit information, including the slope and intercept, were obtained. We performed

different comparisons for both the entire study period and for specific events that we wanted to examine in greater detail.

### 2.5. Meteorological Information

Meteorological measurements including temperature, RH, and wind speed/direction were used from the EPA website (https://www.epa.gov/outdoor-air-quality-data). Only some AQMSs had these meteorological measurements: 42-3-1376-1 and

42-3-8-3 from Pittsburgh, 8-31-26-3 and 8-31-28-3 from Denver, 49-57-2-5 from Ogden-South Ogden, 49-49-4001-5 from Lindon-Orem, and 49-35-3006-4 from Salt Lake City.

Additional meteorological measurements such as temperature, RH, wind speed and gust, wind direction, and visibility of different meteorological stations were obtained from the Iowa Environmental Mesonet website

(https://mesonet.agron.iastate.edu/request/download.phtml). For meteorological information about the selected locations, the following meteorological stations were used: AGC-Pittsburgh/ Allegheny station in Pittsburgh, the Denver International Airport (DEN) station in Denver, the Ogden-Hinckley Muni (OGD) station in Utah, the Provo Muni (PVU) station in Ogden-South Ogden, the Salt Lake City International airport (SLC) station in Lindon-Orem, the California Oakland (OAK) station in Berkeley-Oakland and San Francisco, and the Napa County (APC) station in Vallejo.

### 2.6. AQI Calculations

The AQI is used for the reporting air quality levels. It allows the public to know how clean the air is and indicates the health effects a person may experience within a few hours or days of breathing unhealthy air. The AQI has six categories, each of which corresponds to a different level of health concern (EPA, 2014): Good (0–50, green), Moderate (51–100, yellow),

Unhealthy for Sensitive Groups (101–150, orange), Unhealthy (151–200, red), Very Unhealthy (201–300, purple), and Hazardous (301–500, maroon) (see Table S1). In our study, we calculated the AQI for PM$_{2.5}$ daily average as follows:

$$AQI = \frac{(\text{measured PM}_{2.5} - \text{PM}_{min})(AQI_{max} - AQI_{min})}{(\text{PM}_{max} - \text{PM}_{min})} + AQI_{min} \tag{1}$$

where the measured PM$_{2.5}$ is the daily average PM$_{2.5}$ value, PM$_{max}$ and PM$_{min}$ are respectively the maximum and minimum concentration of the AQI color category for the measured PM$_{2.5}$, AQI$_{max}$ is the maximum AQI value for a color category that

corresponds to the measured PM$_{2.5}$, and AQI$_{min}$ is the minimum AQI value for a color category that corresponds to the measured PM$_{2.5}$. Table S1 lists the different values and categories of PM$_{max}$, PM$_{min}$, AQI$_{max}$, and AQI$_{min}$.

### 3. Result and Discussion



### 3.1. Hourly and Daily PM$_{2.5}$ Comparisons of AQMS and PA-II units.

This study examined measurements for a two-year period from January 1, 2017, to December 31, 2018, resulting in ample overlapping measurement times between the PA-II units and the different AQMSs. The number of concurrent hourly measurements in each comparison varies per location. Overall, the number of concurrent hourly measurements ranged from 1017 to 13975 h with an average of 6652 ± 2822 h per comparison. Other than the Lindon-Orem area where the local AQMS was active only from November 2017, measurements from January 2017 were available in all the other areas. Most of the PA-

II units became active only at the end of 2017. The distance between the different AQMSs and PA-II units ranged from 0.01 km to 13 km with an average of 4.2 ± 2.4 km. Table 2 lists the exact distance and number of PM$_{2.5}$ hourly measurements used in comparisons of each AQMS and PA-II unit. Based on the overlap times, we identified and examined the distribution of daily PM$_{2.5}$ values measured by the PA-II units and AQMS for each location and also performed additional comparisons between the units in these locations.

### 3.1.1 Distribution of Daily PM$_{2.5}$ Values

Fig. 2 shows the distribution of daily PM$_{2.5}$ values for each unit at each of the eight locations. Overall, the daily PM$_{2.5}$ values obtained from both the AQMS and the PA-II units seem to follow similar trends. When the AQMS values increase/decrease, the PA-II values also increase/decrease. The PA-II unit measurements of daily PM$_{2.5}$ values start at 0 µg m$^{-3}$, and the AQMS

can measure negative values owing to its calibration process. In some cases (locations and times), the AQMS measured higher PM$_{2.5}$ daily values compared to the PA-II units, as seen during April–July 2018 in Berkeley-Oakland (Fig. 2C), Lindon-Orem (Fig. 2G), and Salt Lake City (Fig. 2H). However, regardless of the PM$_{2.5}$ concentration, PA-II units usually measured higher values compared to those measured by the AQMS (see July and August 2018 in Pittsburgh, Fig. 2A). This overestimating of PM values by the PA-II units (or PMS sensors) compared to FRM and FEM units has also been observed previously (Kelly et

al., 2017; AQ-SPEC, 2018; Gupta et al., 2018; Sayahi et al., 2019) when the two were co-located.

### 3.1.2 Linear Regression Tests

To evaluate the overall trends of the PA-II units compared to the AQMS, we performed a series of regression tests for each site. As in previous works (Gupta et al., 2018; Sayahi et al., 2019) and as commonly used (Clements et al., 2017), these

225 comparisons were performed using linear regression. Each AQMS was compared to all the PA-II units in its area based on hourly PM$_{2.5}$ measurements. Table 2 lists R$^2$, RMSE values, and the slope and intercept of the linear fit. In general, the linear regression results were mixed. The total R$^2$ values for the hourly PM$_{2.5}$ measurements ranged from 0.1 to 0.91 with an average of 0.63 ± 0.17, which is relatively high. The RMSE values ranged from 3.89 to 13.13 µg m$^{-3}$ with an average of 7.73 ± 2.05 µg m$^{-3}$. The slope ranged from 0.03 to 3.12, but was mostly around 1, with an average of 1.15 ± 0.35.

In some locations such as Denver (Table 2B) and Vallejo (Table 2F), high correlation values were found between the local AQMS and the PA-II units. Denver had three AQMSs; each comparison had a high R$^2$ value in the range of 0.53 to 0.91





(average of 0.72 ± 0.1 for all three AQMSs), average RMSE of 5.65 ± 0.89 µg m$^{-3}$, and average slope of 1.4 ± 0.18. Vallejo had one AQMS with fifteen PA-II units; the R$^2$ values ranged from 0.55 to 0.91 with an average of 0.79 ± 0.13. The RMSE

values in Vallejo were higher than those in Denver, with an average of 8.95 ± 1.28 µg m$^{-3}$ but with lower average slope of 1.27 ± 0.11. These high correlation values and relatively low RMSE indicate that although the PA-II units and the AQMS are not co-located, they still tend to behave in a similar way. At the other locations, except for Ogden-South Ogden, more than 75 % of the comparisons had high correlation values (>0.5) and only a few with low R$^2$ value. Several PA-II units had low R$^2$ values when compared to an AQMS, as in the case of unit 5414 in Berkeley-Oakland and unit 6344 in San Francisco. These two units

also had low correlation values compared to the other PA-II units in their region (data not shown). We noticed that unit 6344 was exposed to very high PM$_{2.5}$ concentrations (up to 250 µg m$^{-3}$ for a duration of 3 h) on May 13, 2018. We suspect that this exposure might have affected the instrument efficiency, as was suggested by Sayahi et al. (2019), and therefore, its measurements differ substantially from those of the AQMS. Another exception was Ogden-South Ogden, as all of the comparisons had very low R$^2$ values (ranging from 0.11 to 0.36 with an average of 0.28 ± 0.1) and high RMSE values (ranging

from 8.27 to 10.6 µg m$^{-3}$). However, when the PA-II units were compared to each other (and not to the AQMS), they showed high correlation values ranging from 0.83 to 0.98 with an average of 0.92 ± 0.05 (Fig. S2). These low correlation values and high RMSE values for the PA-II and AQMS comparisons were most likely caused by specific events and the location of each of the units, as explained below.

A comparison based only on hourly PM$_{2.5}$ values lower than 40 µg m$^{-3}$, as performed by Sayahi et al. (2019), did not improve the hourly correlation values, as shown in Table S2. Around 88 % of the comparisons had lower correlation values compared to the case when all PM$_{2.5}$ concentrations were used; the R$^2$ values ranged from 0.04 to 0.9 with an average of 0.57 ± 0.16. Some locations such as Pittsburgh (Table S2A) showed no change in their correlation values for PM$_{2.5}$ <40 µg m$^{-3}$ comparisons whereas others such as Ogden-South Ogden (Table S2F) and Lindon-Orem (Table S2G) showed improved correlation values.

Unlike the correlation values, the RMSE values in the comparison of PM$_{2.5}$ < 40 µg m$^{-3}$ improved in 93 % of the cases, resulting in lower RMSE values compared to those found when all PM$_{2.5}$ values were used. The RMSE values ranged from 2.89 to 12.96 µg m$^{-3}$ with an average of 6.83 ± 1.54 µg m$^{-3}$.

Comparisons based on the PM$_{2.5}$ daily values improved the results (Table S3). The numbers of concurrent PM$_{2.5}$ daily

measurements ranged from 18 to 574 days, with an average of 270 ± 119 days per comparison. The correlation values ranged from 0.17 to 0.97 with an average of 0.78 ± 0.15. Further, the RMSE values had a wide range of 2.1–12.8 µg m$^{-3}$ with an average of 4.98 ± 1.77 µg m$^{-3}$. Overall, 95 % of the comparisons had a higher R$^2$ and 98 % of the comparisons had lower RMSE values compared to the hourly comparison. Even Ogden-South Ogden, which did not show an improvement in previous comparisons, exhibited better results (Table S3F). The average correlation values in Ogden-South Ogden improved from 0.28

± 0.1 in the hourly comparison to 0.53 ± 0.12 in the daily comparison. The RMSE values also improved; they decreased from an average of 9.51 ± 0.83 µg m$^{-3}$ in the hourly comparisons to 6.95 ± 0.46 µg m$^{-3}$ in the daily comparisons.





## 3.2. Comparison of High Pollution Events

Different meteorological conditions such as wind direction or speed as well as pollution type (traffic, industrial, wildfire,
fireworks, etc.) or source (local vs. regional) may affect the comparison between the AQMS and the PA-II units. We aimed to
determine how the PA-II units behave in a high-pollution event when the daily $PM_{2.5}$ concertation exceeds the EPA daily
regulation of 35 μg m$^{-3}$ Therefore, we decided to investigate specific events with high $PM_{2.5}$ concentrations in different time
frames under different atmospheric conditions.

### 3.2.1. Fireworks in Ogden- South Ogden

In Ogden-South Ogden, major differences were observed in the $PM_{2.5}$ values measured during July 2018 (Fig. 3) by the PA-II
units and the single AQMS. During this month, we noticed that the AQMS measured very high hourly $PM_{2.5}$ values (with
peaks over 400 μg m$^{-3}$), whereas none of the PA-II units exceeded 20 μg m$^{-3}$. The regression test results for this month also
showed low $R^2$ values with an average of $0.03 \pm 0.01$. The location of the units (Fig. S1F), pollution type during this event,
and meteorological conditions at the time revealed the cause of these differences. The increase in $PM_{2.5}$ was due to 4[th] of July
fireworks (correlated to July 5, UTC time) that caused an increase in AQMS hourly $PM_{2.5}$ values > 100 μg m$^{-3}$ for a duration
of 5 h. The AQMS was located downwind from the main fireworks event (Friendship Park, south of the AQMS) whereas all
the PA-II units were far from any fireworks in a residential area on the slopes of Mt. Ogden. Local regulations did not allow
the use of fireworks in a residential area (east of road 203; Ogden City Fire Department, 2019) where most of the PA-II units
are located. Wind direction information obtained from the local metrological station (see Methods) revealed that the wind was
blowing from the fireworks location toward the AQMS but was not reaching the PA-II units. Therefore, the PA-II units could
not detect this increase. A similar result was seen in the previous year in July 2017 when only one PA-II unit was active (see
Fig. 2F). We also noticed that on July 9, one of the PA-II units (ID 6604) measured high $PM_{2.5}$ values (up to 135 μg m$^{-3}$)
whereas all the other units measured much lower $PM_{2.5}$ values. This high concentration was measured during only one hour
(23:00 UTC time); therefore, we suspected that this increase was caused by a local source near this specific unit, such as a
small-scale fire, lawn mower, or barbeque.

In both cases, the presence of the PA-II sensors significantly benefited the areas' residents by allowing them to make informed
decisions. In the case of the fireworks, if the residents were to base their actions solely on the AQMS data, they would assume
that the air quality is unhealthy when actually it is not. If the wind direction was to change and blow from the fireworks toward
the residential area, the AQMS data would not prepare the residents at all. In the second case, the localized pollution was





identified by the PA-II unit; the AQMS did not measure any changes owing to its location. Overall, the probability of any event being identified by a single AQMS is significantly lower than that of it being identified using multiple PA-II sensors.

The remaining days included both low-pollution days (July 1–5 and after July 9) and elevated-pollution days (July 7–8). During these days, the PA-II sensors and the AQMS exhibited similar trends, identified the same changes in $PM_{2.5}$ concentrations, and measured similar values. A repeat of the regression tests for only these days (without the fireworks and local event data) resulted in a significant improvement in correlation values; specifically, the average $R^2$ value increased to $0.69 \pm 0.03$.

**3.2.2. Inversion in Utah**

In Utah, all three locations- Ogden-South Ogden, Lindon-Orem, and Salt Lake City-followed similar daily $PM_{2.5}$ trends during December 4-13, 2018 (Fig. 4). The entire area was affected by an inversion for several days (December 3–13) that increased the daily $PM_{2.5}$ values up to $67.2 \pm 4.17\ \mu g\ m^{-3}$ and reduced the visibility to almost zero (see photos in Williams, 2019). Overall, at each of these three locations, the values measured by the PA-II units increased at the same time and followed a similar trend

to the AQMS measurements. However, whereas all the PA-II units measured similar $PM_{2.5}$ values, the AQMS measured lower $PM_{2.5}$ concentrations. $PM_{2.5}$ values only decreased after precipitation occurred on December 13. The linear regression for each area shows good correlation. In Ogden-South Ogden, Salt Lake City, and Lindon-Orem, the average $R^2$ was $0.93 \pm 0.01$, $0.98 \pm 0.01$ for both AQMSs, and $0.96 \pm 0.01$, respectively. Overall, at each of these three locations, the PA-II units measured similar values, but these seemed to be overestimated when compared to the AQMS measurements.

**3.2.3. Wildfire in California**

The three locations in California- Vallejo, Berkeley-Oakland, and San Francisco are relatively close to each other and were affected by a large wildfire that occurred in November 2018. According to the California Statewide Wildfire Recovery Resources (2019), the wildfire started on November 8 at Butte County (north of Vallejo) owing to a combination of strong

winds and very dry conditions. A southwesterly wind transferred the wildfire smoke from Butte County toward Vallejo, Berkeley-Oakland, and San Francisco. Very high daily $PM_{2.5}$ values ($>200\ \mu g\ m^{-3}$) were measured from November 9 to 21 (Fig. 5). During this period, the area had stable meteorological conditions, with low wind speed, that reduced visibility down to 1.6 km (1 mile). The high daily $PM_{2.5}$ values decreased only after precipitation started on November 21. Overall, at each of the three locations, the values measured by the PA-II units increased at the same time and followed a similar trend to the

AQMS measurements. Regression test results of each area also show very similar results to each other. In Vallejo, the average $R^2$ was $0.97 \pm 0.01$, and in Berkeley-Oakland, where there are three AQMSs, two of them had an average $R^2$ of $0.95 \pm 0.04$ and the third had average $R^2$ of $0.94 \pm 0.03$. In both Vallejo (nine PA-II units) and Berkeley-Oakland (six PA-II units), the average daily $PM_{2.5}$ values of the PA-II units were higher than those measured by the AQMS (Fig. 5A-B). There was no active AQMS at San Francisco during these days, and therefore, only the PA-II units are shown in Fig. 5C. Out of the eight PA-II





units located in Berkeley-Oakland (Fig. 5B), two PA-II units (5414 and 10114) measured lower daily $PM_{2.5}$ values compared to the other PA-II units and even compared to the local AQMS.

Using AQI maps is another good way to see the spatial and temporal changes in $PM_{2.5}$ measurements; it is also important as the public's behavior is based on the interpretation of the AQI values. We calculated the AQI values for both the PA-II units

and the AQMS of all three areas; these calculations were based on the daily $PM_{2.5}$ values (see Methods). We drew maps of all three areas for each day (Fig 6) that show the locations of the AQMS and PA-II units; the locations on the maps are color-coded based on the AQI value at that location on that day. Examining these maps shows us how, as the wildfire and smoke progressed, the air quality worsened. On November 6, before the wildfire started, the AQI for the entire area was moderate. As the fire progressed, the air quality changed from unhealthy on November 11 to very unhealthy on November 16; the air

quality became good again only on November 22. Overall, the AQMS and PA-II units in these areas reported similar values and followed similar trends; AQI values differed between the AQMS and PA-II units on a few days are a result of the differences in the $PM_{2.5}$ values used in the calculation. Having multiple PA-II units in each area allows us to track air quality changes with higher resolution, as multiple sensors provide more data than a single AQMS. In the case of the San Francisco area where no AQMS was active, the PA-II units are the only source of data for providing the residents with crucial information

about the air quality in their region.

### 3.3. Factors That May Impact PA-II Performance

Meteorological conditions such as wind direction and speed, pollutant type, and pollution source are some of the factors that might affect the performance of the PA-II units. It is therefore important to also evaluate and consider additional factors such

as other meteorological conditions and underlying technology used when comparing the behavior and measurements of the PA-II units and the AQMS.

### 3.3.1. Temperature and RH

The sensitivity of the PA-II unit to changes in temperature and RH remains unknown (Gupta et al., 2018). We can assume that

changes in temperature or RH may affect the performance of the PA-II unit especially under atmospheric conditions as they cannot be controlled. Jayaratne et al. (2018) tested an older version of the PMS unit (PMS1003) and reported such an effect. Most low-cost sensors have no heater or dryer at their inlet to remove water from the sample before measuring the particles; therefore, deliquescent or hygroscopic growth of particles, mainly under high RH conditions (>75 %), can lead to higher reported PM concentrations (Jayaratne et al., 2018). According to Rai et al. (2017), most low-cost sensors show some

sensitivity to RH conditions but not to temperature. It is therefore important to evaluate whether the PA-II unit will be affected by changes in temperature or RH. To do so, we used temperature and RH measurements from the nearest available meteorological stations (see Methods for station information) and, in some cases, additional measurements from the AQMS (e.g., in Pittsburgh, Denver, Ogden-South Ogden, Lindon-Orem, and Salt Lake City).





The hourly temperature measurements from the meteorological stations were compared with the hourly $PM_{2.5}$ measurements

from each PA-II unit (86 units in total) using linear regression. The regression resulted in very low $R^2$ values that ranged from $1 \times 10^{-9}$ to 0.07 with an average of $0.02 \pm 0.02$. Similar results were found when the AQMS temperature measurements were used (52 units in total, Table S4); the $R^2$ values ranged from $6 \times 10^{-5}$ to 0.13 with an average of $0.04 \pm 0.03$. For the RH, two different comparisons were made: a comparison using all RH values and a comparison for only those cases in which the RH value was higher than 75 %. When using RH data from the meteorological stations and for the entire RH range, very low $R^2$

values were found. The correlations values ranged from $7.5 \times 10^{-7}$ to 0.1 with an average of $0.02 \pm 0.03$. Comparison results obtained using RH measurements from the AQMS were similar (Table S4); the $R^2$ values ranged from $1.01 \times 10^{-5}$ to 0.17 with an average of $0.05 \pm 0.04$. Even when only RH > 75 % was tested, the $R^2$ values ranged from $1.6 \times 10^{-7}$ to 0.1 with an average of $0.01 \pm 0.01$ for RH measurements from the meteorological station. Similar values were also found for RH measured by the AQMS; $R^2$ values ranged from $5.5 \times 10^{-6}$ to 0.18 with an average of $0.02 \pm 0.04$. Similar results have been reported previously

as well. For example, Sayahi et al. (2019) found very low correlation values between measurements from the PMS5003 sensor and the temperature/RH under atmospheric conditions. Holstius et al. (2014) found a negligible effect of temperature or RH on measurements performed using low-cost sensors under ambient conditions. However, several studies that used old PMS units, such as PMS1003 that was used in PA-I or PMS3003 that was never used in any PA units, found that these sensors were affected by RH (Kelly et al., 2017; Jayaratne et al., 2018; Zheng et al., 2018). AQ-SPEC (2018) tested the PA-II unit in a

laboratory setting under different temperature and RH conditions and found that most temperature and RH combinations had a minimal effect on the PA-II's precision. Our findings for PA-II units in the field under atmospheric conditions are in agreement with those of the AQ-SPEC (2018).

### 3.3.2. Technology, Maintenance, and Placement

There are many differences between PA-II and AQMS units that can influence the comparison results, including the underlying technology and the manner in which units are placed. The $PM_{2.5}$ sensors in the AQMS perform gravimetric measurements using the mass of the particle; by contrast, the PA-II unit uses a laser particle counter to count electric pulses generated as particles cross through a laser beam. Another difference is the physical location of the units; whereas AQMSs are meticulously positioned in an open area, the location of a PA-II sensor is determined by its owner. Although PurpleAir recommends

positioning the PA-II sensor in an open area, ultimately, it is the owner's decision. In practice, most of the PA-II units are located in residential areas with low-rise housing. Further, the height at which the sensor is located could affect the measurements. Whereas the height of the AQMS inlet is regulated and kept constant at each location, the owner of a PA-II unit can freely place it near the ground or higher up. The location of the PA-II units in residential areas can provide both an advantage and a disadvantage. For example, as in the case of Ogden-South Ogden, a single unit might be exposed to more

localized PM sources such as a barbeque, lawn mower, or car, making it report different results compared with other units in its area. Maintenance and calibration are other possible causes of differences between the two. The $PM_{2.5}$ sensors in the AQMS have strict rules for the monthly evaluation of sensor performance, including through flow calibration or calibration based on



minimum value threshold (which, in some cases, causes the recording of negative PM values). By contrast, PA-II units do not have any quality control other than that done by the company for each sensor before shipment to the customer (PurpleAir personal communication, 2019).

### 3.3.3. Distance and Number of Comparisons Between the Units

Other factors that could affect the comparisons with the AQMS are the distances between the units or the number of observations. Previous studies obtained good results when comparing between the PA-II unit or PMS5003 sensor and the FRM and FEM units when the two units were co-located. The AQ-SPEC (2018) recently released a report comparing PA-II units to two FEM instruments under laboratory and field conditions. They found good correlations for hourly and daily values of both $PM_{2.5}$ and $PM_{10}$ under field conditions with higher correlation values for $PM_{2.5}$ compared to those for $PM_{10}$. Gupta et al. (2018) compared three PA-II units in California to a single FEM unit and obtained good correlation values ($R^2 > 0.9$). Sayahi et al. (2019) co-located reference air monitors (tapered element oscillating microbalance, TEOM), and FRM unit, next to a PMS5003 (used in the PA-II unit) in Salt Lake City. The PMS5003 $PM_{2.5}$ measurements correlated well with the hourly TEOM measurements ($R^2 > 0.87$) and with the daily FRM measurements ($R^2 > 0.88$). In our study, we did not position the PA-II units. Further, in most cases, the AQMS and the PA-II units were not located at the same place; therefore, they might have been exposed to different particle types and concentrations. Some might claim that not having the PA-II and FRM units co-located, as was done in previous studies, might diminish the accuracy of the comparison between these units. Although lower correlation values were in fact observed in our study, as we were using PA-II units in their natural locations, this was expected. Further, as we saw that the correlation values are not much lower than those in the co-located cases described in previous studies, they are still statistically significant. Because the AQMS and the PA-II units were not co-located, we wanted to verify whether the distance between the AQMS and the PA-II units affected the $R^2$ values. We compared the $R^2$ values that we previously calculated for the hourly $PM_{2.5}$ measurements with the corresponding distances between the PA-II units and AQMS (Fig. S3A). There was no correlation between the two, and similar results were found when the RMSE values were tested (Fig. S3B). The number of observations used for the comparison was also tested; comparing the same $R^2$ from the measurements with the number of observations revealed no effect of the number of observations on $R^2$ or RMSE values (Fig. S3C-D).

### 3.4. Next Steps with PA-II units

Ford et al. (2019) suggested the use of PA-II units as a network installed by residents in an in North Colorado. This seems like a good solution for locations that are lacking FRM or FEM units as multiple sensors can provide more data. However, it is important to consider the limitations of the PA-II unit. The PA-II unit needs to be monitored for changes in unit behavior. We recommend PurpleAir to monitor the measurements of the PA-II units, identify units that behave differently from other surrounding units or units whose internal sensors (A and B) report different values, flag them on the online map, and communicate instructions to the unit owners on how to clean the unit. The manufacturer of the PMS5003 sensor that is used in the PA-II units noted that it has a lifetime of ~3 years (Yong, 2016). None of the current units have been active for that long;



therefore, the efficiency of PA-II units over such a long period remains unknown and should be evaluated. It is possible that, after this duration, they will lose their efficiency (a behavior known as drift) and will become outliers.

**4. Conclusions**

PA-II units are becoming a common low-cost tool to monitor changes in the concentrations of PMs of various sizes. Previous studies have examined the performance of these PA-II units (or the sensor in them) by comparing them with a co-located EPA AQMS. However, PA-II units are not co-located in practice, and some of them are placed in areas where there is no reference air monitor system. This study aimed to examine the behavior of PA-II units under atmospheric conditions when exposed to a

440 variety of pollutants and different $PM_{2.5}$ concentrations. For this purpose, we used PA-II units that have already been active for some time irrespective of where they might be. Eight locations with multiple PA-II units and at least a single AQMS were identified. Each PA-II unit was compared to the AQMS and to other PA-II units in its surrounding area based on hourly or daily $PM_{2.5}$ measurements. Overall, the PA-II units behaved in a similar way to the other PA-II units at their locations. We found that even though some PA-II units overestimated or underestimated at times, the AQMS and PA-II units were mostly in

agreement and measured similar $PM_{2.5}$ concentrations. PA-II was also found to not be affected by temperature or RH. We think that the PA-II unit is a promising tool for measuring $PM_{2.5}$ concentrations and identifying relative concentration changes. Further, through the use of AQI, the current air quality can be successfully conveyed to the public. The PA-II unit has the potential to complement sparsely distributed monitoring stations, particularly in areas lacking a nearby AQMS.

**Data availability.** All data can be provided by the authors upon request.

**Competing interests.** The authors declare that they have no conflict of interest.

**Acknowledgment**

The authors are thankful to the PurpleAir team for their help and explanations about the PA-II units. Further, they are thankful to Mr. Mangus Nick from the National Air Data Group at US EPA for his help with the EPA data. Finally, they are thankful to Dr. Amber McCord, College of Media & Communication at Texas Tech University, for her help with the graphic abstract. Use of the sensor manufacturer's name does not imply endorsement.

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





**Table legends**

**Table 1**. Information on each of the eight locations with the names of the AQMS and PA-II units, their location (latitude and
longitude), first and last time of measurement, minimum, and maximum PM$_{2.5}$ hourly values. AQMS ID represented by the
numbers of State-County-Site-POC for each unit.

**Table 2**. Comparison between each AQMS and the different PA-II units per location (A-G) for average hourly PM$_{2.5}$
measurements. Distance and number of observations (hours) are provided for each comparison along with linear regression
result such as R$^2$, RMSE values, and the slope and intercept of the linear fit. Bold R$^2$ values represent values larger than 0.5.

**Figure legends**

**Figure 1**. (A) Picture from the bottom of the PA-II unit containing two PMS5003 sensors (in blue). (B) Schematic of a single
PMS5003 sensor. A fan draws the particles through the inflow (rounded holes) at the lower level of the sensor. The particles
travel to the upper part of the sensor where they come out through the air flow holes and then pass through the laser path,
causing the beam to scatter. Finally, the particles exit from the fan.

**Figure 2**. Distribution of daily PM$_{2.5}$ measurements from the AQMS and PA-II units in each of the eight areas: (A) Pittsburgh;
(B) Denver; (C) Berkeley-Oakland; (D) San Francisco; (E) Vallejo; (F) Ogden-South Ogden; (G) Lindon-Orem, and (H) Salt
Lake City. Measurements from AQMS are represented by the green lines and the PA-II units are indicated by purple lines.
The numbers are the units' ID numbers.

**Figure 3**. Hourly PM$_{2.5}$ measurements at Ogden-South Ogden in UT during July 1-11, 2018 (UTC time). Measurements from
the AMQS unit are represented in green and those from the PA-II units, in different shades of purple. Each number represent
the ID of the unit. Error bars represent the standard deviation values for each hour on each of the PA-II units. Note that local
PA-II unit 465 was not active during this time.

**Figure 4**. Hourly measurements of PM$_{2.5}$ at (A) Ogden-South Ogden, (B) Lindon-Orem, and (C) Salt Lake City during
December 1-14 2018 (UTC time). An increase in average daily PM$_{2.5}$ values was observed from December 4-13. The AMQS
unit is represented by the different green lines and the PA-II units, by the different purple lines. Each number represents the
ID of the unit. Bars represent the standard deviation values per day. Several PA-II units were not operating during these times.

**Figure 5**. Hourly measurements of PM$_{2.5}$ at (A) Vallejo, (B) Berkeley-Oakland (B), and (C) San Francisco during the
November 2018 wildfire (UTC time). An increase in average daily PM$_{2.5}$ values was observed during November 9–20. The
AMQS unit is represented by the different green lines and the PA-II units, by the different purple lines. Each number represent
the ID of the unit. Bars represent the standard deviation values per day.

**Figure 6**. Spatial and temporal changes of AQI in California at Berkeley-Oakland, San Francisco, and Vallejo during
November 8-22, 2018. Squares represent AQMS and circles, PA-II units. The colors of units represent the different AQI
values.



**Table 1**. Information on each of the eight locations with the names of the AQMS and PA-II units, their location (latitude and longitude), first and last time of measurement, minimum, and maximum PM$_{2.5}$ hourly values. AQMS ID represented by the numbers of State-County-Site-POC for each unit.

| Location | Unit Type | ID of Each Unit (PA-II - sensor A) | Latitude | Longitude | PA-II Unit label | First day of observation | Last day of observation | Minimum PM$_{2.5}$ hourly average ($\mu g/m^3$) | Maximum PM$_{2.5}$ hourly average ($\mu g/m^3$) | Number of observations (hours) |
|---|---|---|---|---|---|---|---|---|---|---|
| **1. Pittsburgh, PA** | AQMS | 42-3-8-3 * | 40.465 | -79.961 | | 1-Jan-17 | 31-Dec-18 | -2 | 109 | 17302 |
| | | 42-3-1376-1 & | 40.437 | -79.864 | | 1-Jan-17 | 31-Dec-18 | -2 | 67 | 16690 |
| | PurpleAir sensor ID | 3723 | 40.448 | -79.916 | Point Breeze | 14-Oct-17 | 28-Sep-18 | 0.1 | 86.66 | 3438 |
| | | 3981 | 40.438 | -79.956 | CMU CAPS PPA 010 | 20-Nov-17 | 8-Oct-18 | 0.19 | 80.84 | 2143 |
| | | 9016 | 40.421 | -79.914 | Parkview Blvd-Summerset at Frick Park | 27-May-18 | 31-Dec-18 | 0.36 | 79.69 | 2885 |
| | | 9026 | 40.478 | -79.93 | Jancey St Morningside | 22-Apr-18 | 31-Dec-18 | 0.13 | 47.59 | 3412 |
| | | 9038 | 40.445 | -79.915 | Pillars In Squirrel Hill North | 7-May-18 | 31-Dec-18 | 0.11 | 193.46 | 4250 |
| | | 9906 | 40.436 | -79.908 | Frick Environmental Center - Squirrel Hill | 9-May-18 | 31-Dec-18 | 0.01 | 55.4 | 3499 |
| | | 9878 | 40.45 | -79.911 | juniata ct | 6-May-18 | 31-Dec-18 | 0.09 | 49.39 | 4424 |
| | | 9880 | 40.473 | -79.914 | HP Winterton St | 6-May-18 | 26-Dec-18 | 0.09 | 192.89 | 3957 |
| | | 9892 | 40.43 | -79.918 | Nicholson St | 2-May-18 | 31-Dec-18 | 0 | 116.85 | 5729 |
| | | 9896 | 40.441 | -79.896 | EastEndAve1 | 2-May-18 | 31-Dec-18 | 0.12 | 137.42 | 3378 |
| | | 9906 | 40.43 | -79.954 | South Oakland | 9-May-18 | 31-Dec-18 | 0.03 | 174.57 | 5613 |
| **2. Denver, CO** | AQMS | 8-31-26-3 $ | 39.779 | -105.005 | | 1-Jan-17 | 31-Dec-18 | 0 | 76.5 | 16850 |
| | | 8-31-27-3 # | 39.732 | -105.015 | | 1-Jan-17 | 31-Dec-18 | 0.2 | 73.1 | 17259 |
| | | 8-31-28-3 # | 39.786 | -104.989 | | 1-Jan-17 | 24-Dec-18 | 0.3 | 75.1 | 16651 |
| | PurpleAir sensor ID | 2249 | 39.783 | -104.96 | The GrowHaus | 5-Dec-17 | 31-Dec-18 | 0.08 | 121.99 | 9331 |
| | | 2267 | 39.779 | -105.006 | La Casa | 22-Aug-17 | 27-Feb-18 | 0.08 | 132.44 | 2156 |
| | | 2269 | 39.781 | -104.956 | Swansea (DEH) | 4-Aug-17 | 18-Jun-18 | 0.04 | 170.64 | 7128 |
| | | 2719 | 39.755 | -104.966 | 26th and Williams | 12-Aug-17 | 1-Nov-18 | 0.1 | 155.27 | 6411 |
| | | 2900 | 39.753 | -105.041 | West Denver PA-II | 18-Aug-17 | 31-Dec-18 | 0.04 | 152.73 | 11980 |
| | | 3924 | 39.779 | -105.005 | APCD La Casa | 16-Nov-17 | 31-Dec-18 | 0.05 | 81.58 | 9004 |
| | | 4022 | 39.708 | -104.981 | Wash Park West | 8-Nov-17 | 31-Dec-18 | 0.04 | 80.46 | 9968 |



| | | | | | | | | | |
|---|---|---|---|---|---|---|---|---|---|
| | | 7956 | 39.786 | -104.989 | Globeville | 27-Feb-18 | 31-Dec-18 | 0.11 | 78.48 | 7326 |
| **3. Berkeley -Oakland, CA** | AQMS | 6-1-11-3 * | 37.815 | -122.282 | | 1-Jan-17 | 31-Dec-18 | -10 | 210 | 17210 |
| | | 6-1-12-3 * | 37.794 | -122.263 | | 1-Jan-17 | 31-Dec-18 | -3 | 218 | 17283 |
| | | 6-1-13-3 * | 37.865 | -122.303 | | 1-Jan-17 | 31-Dec-18 | -7 | 393 | 16882 |
| | PurpleAir sensor ID | 2574 | 37.901 | -122.286 | Berkeley Park and Coventry, Kensington, CA, USA | 19-Sep-17 | 31-Dec-18 | 0.05 | 281.12 | 10476 |
| | | 3082 | 37.906 | -122.302 | El Cerrito - Rust - Ohlone Greenway | 6-Sep-17 | 16-Nov-18 | 0.04 | 291.35 | 10176 |
| | | 3854 | 37.862 | -122.247 | Claremont Blvd | 17-Oct-17 | 7-Oct-18 | 0.03 | 87.47 | 8007 |
| | | 4335 | 37.81 | -122.298 | West Oakland, Oakland, CA | 30-Nov-17 | 31-Dec-18 | 0.09 | 239.01 | 9471 |
| | | 4506 | 37.875 | -122.271 | North Berkeley | 3-Dec-17 | 31-Dec-18 | 0.05 | 307.69 | 9434 |
| | | 4795 | 37.797 | -122.216 | Lodestar | 6-Dec-17 | 19-Jun-18 | 0.12 | 58.92 | 4125 |
| | | 4825 | 37.7637 | -122.233 | Northwood | 22-Dec-17 | 31-Dec-18 | 0.08 | 272.35 | 8733 |
| | | 5414 | 37.8295 | -122.248 | Piedmont Ave | 17-Nov-18 | 31-Dec-18 | 0.02 | 211.77 | 1048 |
| | | 6410 | 37.858 | -122.284 | San Pablo Park / The Derby | 15-Mar-18 | 31-Dec-18 | 0.03 | 297 | 6911 |
| | | 10114 | 37.8 | -122.249 | CCEEB - Park & E. 19th | 30-May-18 | 31-Dec-18 | 0.11 | 137.76 | 3927 |
| **4. San Francisco, CA** | AQMS | 6-75-5-3 * | 37.766 | -122.399 | | 1-Jan-17 | 31-Dec-18 | -10 | 190 | 16309 |
| | PurpleAir sensor ID | 1226 | 37.768 | -122.402 | Volta Charging | 17-Oct-17 | 31-Dec-18 | 0.08 | 263.78 | 10417 |
| | | 2031 | 37.733 | -122.424 | St Mary's Park | 15-Sep-17 | 31-Dec-18 | 0.07 | 265.08 | 11295 |
| | | 2910 | 37.778 | -122.408 | tactrix rooftop | 18-Sep-17 | 31-Dec-18 | 0.12 | 282.73 | 10861 |
| | | 3348 | 37.787 | -122.445 | Lower Pacific Heights | 13-Nov-17 | 23-Dec-18 | 0.1 | 180.53 | 3937 |
| | | 3996 | 37.789 | -122.391 | South Beach | 11-Nov-17 | 1-Oct-18 | 0.1 | 79.63 | 7783 |
| | | 4372 | 37.754 | -122.412 | The Mission- Clean air is hip | 5-Jan-18 | 31-Dec-18 | 0.09 | 250.54 | 7951 |
| | | 4770 | 37.787 | -122.417 | 930 Post | 21-Dec-17 | 31-Dec-18 | 0.22 | 250.18 | 8883 |
| | | 5776 | 37.745 | -122.421 | La Lengua Air Station Alpha | 5-Jan-18 | 23-Dec-18 | 0 | 275.6 | 8033 |
| | | 6344 | 37.759 | -122.403 | Kansas Gulch | 28-Jan-18 | 17-Jun-18 | 0.11 | 252.71 | 3384 |
| **5. Vallejo, CA** | AQMS | 6-95-4-4 * | 38.1 | -122.24 | | 1-Jan-17 | 31-Dec-18 | -10 | 435 | 16630 |
| | PurpleAir sensor ID | 1142 | 38.104 | -122.258 | Carolina Street | 14-Apr-17 | 9-Oct-18 | 0.06 | 457.06 | 9893 |
| | | 1870 | 38.111 | -122.243 | Amador St @ Stutz Alley | 17-Jul-17 | 31-Dec-18 | 0.04 | 468.49 | 12646 |
| | | 1874 | 38.067 | -122.22 | Glen Cove Ridge | 15-Jul-17 | 31-Dec-18 | 0.05 | 292.45 | 12460 |
| | | 1878 | 38.086 | -122.245 | Winchester Hill | 20-Jul-17 | 3-May-18 | 0.08 | 384.76 | 6432 |
| | | 1882 | 38.078 | -122.23 | Navone St. | 19-Jul-17 | 26-Dec-18 | 0.05 | 339.3 | 11406 |
| | | 2480 | 38.122 | -122.233 | Howard Ave | 17-Aug-17 | 31-Dec-18 | 0.06 | 477.83 | 11908 |
| | | 2906 | 38.074 | -122.24 | Sandy Beach | 10-Dec-17 | 31-Dec-18 | 0.04 | 303.5 | 9245 |
| | | 3686 | 38.074 | -122.231 | Carquinez One | 6-Dec-17 | 23-Aug-18 | 0.08 | 256.68 | 7143 |
| | | 3758 | 38.114 | -122.259 | Buckles St | 11-Nov-17 | 22-Aug-18 | 0.12 | 92.45 | 6774 |





| | | | | | | | | | |
|---|---|---|---|---|---|---|---|---|---|
| | | 3769 | 38.081 | -122.215 | Old Glen Cove | 14-Oct-17 | 31-Dec-18 | 0.08 | 287.42 | 10426 |
| | | 3782 | 38.12 | -122.241 | El Camino Real/Valle Vista | 29-Nov-17 | 1-Oct-18 | 0.05 | 85.32 | 7401 |
| | | 3784 | 38.098 | -122.26 | Little Old Lady By The River | 20-Oct-17 | 31-Dec-18 | 0.05 | 278.12 | 10148 |
| | | 3960 | 38.141 | -122.26 | 211 Sonora pass rd | 18-Jan-18 | 24-Oct-18 | 0.03 | 227.05 | 6508 |
| | | 4928 | 38.09 | -122.239 | 1300 Block Lemon | 1-Dec-17 | 31-Dec-18 | 0.03 | 296.62 | 8253 |
| | | 5127 | 38.108 | -122.256 | Vallejo | 2-Dec-17 | 31-Dec-18 | 0.08 | 243.68 | 9406 |
| **6. Ogden - South Ogden, UT** | AQMS | 49-57-2-5 ** | 41.21 | -111.98 | | 4-Jan-17 | 31-Dec-18 | -10 | 790.3 | 13574 |
| | PurpleAir sensor ID | 465 | 41.185 | -111.935 | Beus Park | 1-Jan-17 | 30-Nov-17 | 0.06 | 83.72 | 8002 |
| | | 1104 | 41.179 | -111.946 | University Village - Weber State University | 31-Jan-18 | 31-Dec-18 | 0.07 | 96.44 | 7712 |
| | | 5178 | 41.216 | -111.931 | Taylor Canyon | 9-Dec-17 | 31-Dec-18 | 0.57 | 110.39 | 7797 |
| | | 5454 | 41.192 | -111.942 | WSU Marriott Health | 4-Apr-18 | 31-Dec-18 | 0.06 | 64.49 | 5124 |
| | | 6604 | 41.185 | -111.938 | Bobwhite Ct | 1-Feb-18 | 31-Dec-18 | 0 | 135.89 | 5687 |
| | | 7858 | 41.195 | -111.947 | WSU Public Safety Building | 5-Apr-18 | 31-Dec-18 | 0 | 104.43 | 6135 |
| | | 7860 | 41.193 | -111.943 | WSU Stewart Library | 4-Apr-18 | 31-Dec-18 | 0 | 95.09 | 6248 |
| **7. Lindon - Orem, UT** | AQMS | 49-49-4001-5 ** | 40.341 | -111.714 | | 8-Nov-17 | 31-Dec-18 | 0.1 | 204 | 9984 |
| | PurpleAir sensor ID | 5135 | 40.324 | -111.715 | Orem Bonneville Park powered by UTOPIA Fiber | 10-Jan-18 | 31-Dec-18 | 0.08 | 165.33 | 8420 |
| | | 5143 | 40.315 | -111.667 | Orem Foothill Park powered by UTOPIA Fiber | 19-Dec-17 | 19-Oct-18 | 0.01 | 53.67 | 3949 |
| | | 5145 | 40.308 | -111.705 | Orem 600N 400W powered by UTOPIA Fiber | 18-Jan-18 | 17-Jun-18 | 0.12 | 46.11 | 3471 |
| | | 5728 | 40.314 | -111.697 | Orem Fire Department #2 powered by UTOPIA Fiber | 19-Jan-18 | 31-Dec-18 | 0 | 166.86 | 7273 |
| | | 5732 | 40.308 | -111.73 | Orem Public Works powered by UTOPIA Fiber | 18-Jan-18 | 31-Dec-18 | 0 | 192.96 | 7440 |
| | | 5736 | 40.299 | -111.705 | Orem 400W 75N powered by UTOPIA Fiber | 18-Jan-18 | 31-Dec-18 | 0 | 183.61 | 7315 |
| | | 5750 | 40.302 | -111.712 | Orem Geneva Park powered by UTOPIA Fiber | 21-Jan-18 | 19-Oct-18 | 0 | 187.95 | 6253 |
| | | 5754 | 40.317 | -111.677 | Orem Orchard Elementary powered by UTOPIA Fiber | 19-Jan-18 | 19-Oct-18 | 0 | 102.15 | 6089 |
| | | 5760 | 40.31 | -111.713 | Orem Junior High powered by UTOPIA Fiber | 23-Oct-18 | 31-Dec-18 | 0.09 | 72.01 | 1628 |
| | | 6304 | 40.308 | -111.689 | Orem Sharon Park powered by UTOPIA Fiber | 1-Feb-18 | 31-Dec-18 | 0 | 130.03 | 7882 |
| | | 6948 | 40.338 | -111.694 | Lindon City - Murdock Canal Trail | 21-Mar-18 | 24-Aug-18 | 0.17 | 100.75 | 2987 |
| | | 6986 | 40.34 | -111.718 | Lindon City Center | 20-Mar-18 | 31-Dec-18 | 0 | 156.33 | 4954 |
| **8. Sal** | AQMS | 49-35-3006-4 ** | 40.74 | -111.87 | | 1-Jan-17 | 31-Dec-18 | 0 | 87.5 | 16529 |



| | | | | | | | | | |
|---|---|---|---|---|---|---|---|---|---|
| | 49-35-3006-5 ** | 40.74 | -111.87 | | | 1-Jan-17 | 31-Dec-18 | -10 | 89.1 | 17030 |
| PurpleAir sensor ID | 884 | 40.777 | -111.895 | Quince and Apricot | 15-Feb-17 | 31-Dec-18 | 0 | 114.33 | 14426 |
| | 3388 | 40.733 | -111.822 | Montessori Community School | 20-Oct-17 | 31-Dec-18 | 0.03 | 79.22 | 10248 |
| | 5014 | 40.771 | -111.9 | KSL Triad | 28-Nov-17 | 31-Dec-18 | 0.08 | 123.65 | 9520 |
| | 5460 | 40.728 | -111.861 | 1027 Hollywood | 14-Jan-18 | 31-Dec-18 | 0.08 | 125.71 | 8097 |
| | 5742 | 40.734 | -111.846 | Wasatch Hollow | 7-Jan-18 | 31-Dec-18 | 0 | 156.33 | 6815 |
| | 5802 | 40.71 | -111.832 | Yuma View | 7-Jan-18 | 14-May-18 | 0.03 | 47.05 | 2610 |
| | 5990 | 40.72 | -111.82 | Lynwood | 5-Jul-18 | 31-Dec-18 | 0.04 | 187.18 | 4013 |
| | 6078 | 40.764 | -111.86 | Victory Park | 29-Jan-18 | 25-Jul-18 | 0.03 | 39.9 | 1231 |
| | 6356 | 40.774 | -111.883 | Cobble Knoll | 29-Jan-18 | 31-Dec-18 | 0 | 116.68 | 8050 |
| | 6360 | 40.767 | -111.867 | Capitol Hill Construction | 26-Jan-18 | 31-Dec-18 | 0.03 | 92.98 | 8105 |
| | 6434 | 40.696 | -111.877 | 3450 South 500 East | 26-Jan-18 | 31-Dec-18 | 0.08 | 157.06 | 7875 |
| | 6608 | 40.774 | -111.851 | 4th AveCat | 5-Mar-18 | 31-Dec-18 | 0 | 243.54 | 7179 |
| | 6622 | 40.744 | -111.876 | Tracy Aviary | 25-Feb-18 | 31-Dec-18 | 0 | 128.51 | 6468 |
| | 10050 | 40.749 | -111.912 | Utah Paperbox | 2-May-18 | 31-Dec-18 | 0.15 | 150.72 | 5771 |

**AQMS Sensor Type - *** Met One BAM-1020 Mass Monitor; ** Thermo Scientific Model 5030; & Thermo Scientific 5014i; $ Teledyne T640; # GRIMM EDM Model 180





**Table 2**. Comparison between each AQMS and the different PA-II units per location (A-G) for average hourly $PM_{2.5}$ measurements. Distance and number of observations (hours) are provided for each comparison along with linear regression result such as $R^2$, RMSE values, and the slope and intercept of the linear fit. Bold $R^2$ values represent values larger than 0.5.

| A. Pittsburgh | | | PurpleAir sensor ID | | | | | | | | | | |
|---|---|---|---|---|---|---|---|---|---|---|---|---|---|
| | | | 3723 | 3981 | 9016 | 9026 | 9038 | 9096 | 9878 | 9880 | 9892 | 9896 | 9906 |
| **EPA AQMS ID** | **42-3-1376-1** | Distance (km) | 4.58 | 7.79 | 4.66 | 7.24 | 4.44 | 3.73 | 4.24 | 5.79 | 4.65 | 2.79 | 7.72 |
| | | Obs (h) | 3394 | 2116 | 2861 | 3380 | 4207 | 3470 | 4379 | 3913 | 5672 | 3352 | 5558 |
| | | $R^2$ | **0.51** | 0.43 | **0.54** | **0.53** | **0.57** | **0.61** | **0.59** | **0.51** | **0.57** | **0.54** | 0.49 |
| | | RMSE | 8.04 | 8.72 | 7.35 | 6.42 | 7.22 | 6.49 | 6.35 | 7.50 | 6.90 | 7.17 | 7.63 |
| | | Slop | 0.99 | 0.86 | 1.10 | 0.99 | 1.16 | 1.16 | 1.09 | 1.06 | 1.16 | 1.12 | 1.06 |
| | | Intercept | 4.29 | 6.18 | 3.61 | 4.23 | 2.37 | 2.01 | 2.72 | 3.05 | 2.33 | 3.93 | 3.43 |
| | **42-3-8-3** | Distance (km) | 4.26 | 3.09 | 6.32 | 2.96 | 4.48 | 5.54 | 4.55 | 4.07 | 5.34 | 6.1 | 4 |
| | | Obs (h) | 3207 | 2035 | 2737 | 3186 | 4026 | 3301 | 4132 | 3677 | 5418 | 3128 | 5300 |
| | | $R^2$ | **0.52** | **0.51** | 0.46 | **0.58** | **0.56** | 0.49 | **0.57** | **0.50** | **0.52** | 0.46 | **0.53** |
| | | RMSE | 8.04 | 8.12 | 8.08 | 6.21 | 7.39 | 7.48 | 6.60 | 7.63 | 7.37 | 7.91 | 7.37 |
| | | Slop | 1.20 | 1.18 | 1.11 | 1.09 | 1.26 | 1.13 | 1.16 | 1.16 | 1.20 | 1.10 | 1.21 |
| | | Intercept | 2.91 | 3.25 | 0.34 | 0.14 | -1.76 | -0.54 | -1.03 | -0.99 | -0.69 | 1.20 | -0.84 |

| B. Denver | | | PurpleAir sensor ID | | | | | | | |
|---|---|---|---|---|---|---|---|---|---|---|
| | | | 2249 | 2267 | 2269 | 2719 | 2900 | 3924 | 4022 | 7956 |
| **EPA AQMS ID** | **8-31-26-3** | Distance (km) | 3.89 | 0.08 | 4.25 | 4.34 | 4.23 | 0.01 | 8.19 | 1.57 |
| | | Obs (h) | 9130 | 2144 | 7060 | 6336 | 11763 | 8807 | 9765 | 7151 |
| | | $R^2$ | **0.76** | **0.91** | **0.81** | **0.81** | **0.80** | **0.81** | **0.73** | **0.75** |
| | | RMSE | 4.80 | 4.26 | 5.01 | 5.11 | 4.79 | 4.51 | 5.15 | 4.36 |
| | | Slop | 1.41 | 1.70 | 1.52 | 1.51 | 1.50 | 1.54 | 1.37 | 1.40 |
| | | Intercept | -1.25 | -2.25 | -2.21 | -2.41 | -1.61 | -1.77 | -1.45 | -0.74 |
| | **8-31-27-3** | Distance (km) | 7.34 | 5.26 | 7.49 | 4.93 | 3.19 | 5.33 | 3.95 | 6.39 |
| | | Obs (h) | 8708 | 2145 | 6859 | 6319 | 11338 | 8407 | 9337 | 6907 |
| | | $R^2$ | **0.67** | **0.83** | **0.74** | **0.75** | **0.73** | **0.70** | **0.70** | **0.68** |
| | | RMSE | 5.64 | 6.04 | 5.91 | 5.91 | 5.60 | 5.75 | 5.45 | 4.85 |
| | | Slop | 1.37 | 1.64 | 1.51 | 1.49 | 1.47 | 1.47 | 1.38 | 1.33 |
| | | Intercept | -1.80 | -2.73 | -2.92 | -3.27 | -2.24 | -2.24 | -2.43 | -1.01 |





|  |  |  |  |  |  |  |  |  |
|---|---|---|---|---|---|---|---|---|
| **8-31-28-3** | Distance (km) | 2.49 | 1.66 | 2.87 | 3.99 | 5.78 | 1.59 | 8.68 | 0.03 |
|  | Obs (h) | 8750 | 2145 | 6970 | 5956 | 11380 | 8444 | 9382 | 6866 |
|  | $R^2$ | **0.61** | **0.78** | **0.65** | **0.62** | **0.59** | **0.59** | **0.53** | **0.66** |
|  | RMSE | 6.15 | 6.83 | 6.80 | 7.29 | 6.97 | 6.67 | 6.75 | 5.00 |
|  | Slop | 1.11 | 1.72 | 1.40 | 1.35 | 1.19 | 1.15 | 1.04 | 1.07 |
|  | Intercept | -0.86 | -4.34 | -2.72 | -2.94 | -1.07 | -0.79 | -0.76 | -0.03 |

| **C. Berkeley -Oakland** |  |  | PurpleAir sensor ID |  |  |  |  |  |  |  |  |
|---|---|---|---|---|---|---|---|---|---|---|---|
|  |  |  | 2574 | 3082 | 3854 | 4335 | 4506 | 4795 | 4825 | 5414 | 6410 | 10114 |
| EPA AQMS ID | **6-1-11-3** | Distance (km) | 9.56 | 10.33 | 6.13 | 1.45 | 6.79 | 6.13 | 7.14 | 2.67 | 4.83 | 3.36 |
|  |  | Obs (h) | 10448 | 10147 | 7988 | 9459 | 9422 | 4117 | 8725 | 1046 | 6905 | 3924 |
|  |  | $R^2$ | **0.76** | **0.69** | 0.36 | **0.86** | **0.79** | 0.43 | **0.85** | 0.38 | **0.82** | **0.65** |
|  |  | RMSE | 12.05 | 11.55 | 8.08 | 8.55 | 11.82 | 7.72 | 10.40 | 13.13 | 12.16 | 10.93 |
|  |  | Slop | 1.21 | 1.21 | 0.68 | 1.19 | 1.25 | 0.71 | 1.36 | 0.40 | 1.30 | 0.60 |
|  |  | Intercept | -4.17 | -3.22 | 0.68 | -3.10 | -4.23 | -0.99 | -4.69 | 2.92 | -2.67 | 6.08 |
|  | **6-1-12-3** | Distance (km) | 12.08 | 12.99 | 7.76 | 3.51 | 9.09 | 4.16 | 4.26 | 3.3 | 7.41 | 1.4 |
|  |  | Obs (h) | 10323 | 10026 | 7943 | 9324 | 9287 | 4091 | 8592 | 1042 | 6790 | 3898 |
|  |  | $R^2$ | **0.84** | **0.78** | **0.57** | **0.87** | **0.87** | **0.59** | **0.90** | 0.39 | **0.88** | **0.70** |
|  |  | RMSE | 9.95 | 9.75 | 6.60 | 8.17 | 9.11 | 6.56 | 8.60 | 13.11 | 9.86 | 10.26 |
|  |  | Slop | 1.28 | 1.30 | 0.96 | 1.22 | 1.34 | 0.95 | 1.42 | 0.41 | 1.36 | 0.62 |
|  |  | Intercept | -5.62 | -5.08 | -3.24 | -3.60 | -5.63 | -3.63 | -5.75 | 3.01 | -3.90 | 5.26 |
|  | **6-1-13-3** | Distance (km) | 4.26 | 4.64 | 4.93 | 6.12 | 3.04 | 10.68 | 12.8 | 6.41 | 1.78 | 8.64 |
|  |  | Obs (h) | 10181 | 9912 | 7825 | 9167 | 9114 | 4036 | 8444 | 1017 | 6675 | 3733 |
|  |  | $R^2$ | **0.79** | **0.71** | 0.35 | **0.81** | **0.83** | 0.53 | **0.82** | 0.41 | **0.85** | **0.63** |
|  |  | RMSE | 11.45 | 11.40 | 8.18 | 9.97 | 10.57 | 6.96 | 11.52 | 12.70 | 11.38 | 11.49 |
|  |  | Slop | 1.22 | 1.20 | 0.67 | 1.15 | 1.28 | 0.92 | 1.32 | 0.43 | 1.30 | 0.57 |
|  |  | Intercept | -0.79 | 0.68 | 2.83 | 0.60 | -1.17 | -0.61 | -0.82 | 1.79 | -0.02 | 7.21 |

| **D. San Francisco** |  |  | PurpleAir sensor ID |  |  |  |  |  |  |  |  |
|---|---|---|---|---|---|---|---|---|---|---|---|
|  |  |  | 1226 | 2031 | 2910 | 3348 | 3996 | 4372 | 4770 | 5776 | 6344 |
| EPA AQM | 6-75-3 | Distance (km) | 0.35 | 4.23 | 1.6 | 4.65 | 2.65 | 1.75 | 2.86 | 3.03 | 0.85 |
|  |  | Obs (h) | 9290 | 10157 | 9725 | 3546 | 7558 | 6954 | 7867 | 7024 | 3223 |





| | | | | | | | | | |
|---|---|---|---|---|---|---|---|---|---|
| R² | **0.63** | **0.65** | **0.65** | **0.64** | **0.58** | **0.53** | **0.55** | 0.42 | 0.19 |
| RMSE | 6.71 | 7.58 | 7.82 | 7.30 | 7.34 | 7.16 | 7.15 | 6.97 | 7.41 |
| Slop | 1.03 | 1.07 | 1.11 | 1.01 | 0.98 | 0.97 | 0.96 | 0.76 | 0.53 |
| Intercept | 0.72 | 0.74 | 1.45 | 0.55 | 2.18 | 1.27 | 3.43 | 1.61 | 2.55 |

| **E. Vallejo** | | | PurpleAir sensor ID | | | | | | | | | | | | | | |
|---|---|---|---|---|---|---|---|---|---|---|---|---|---|---|---|---|---|
| | | | 1142 | 1870 | 1874 | 1878 | 1882 | 2480 | 2906 | 3686 | 3758 | 3769 | 3782 | 3784 | 3960 | 4928 | 5127 |
| EPA AQMS ID | 6-95-4-4 | Distance (km) | 1.78 | 1.07 | 4.27 | 1.96 | 2.8 | 2.21 | 3.22 | 3.23 | 2.24 | 3.16 | 2 | 1.96 | 4.73 | 1.35 | 1.64 |
| | | Obs (h) | 9525 | 11824 | 11647 | 6257 | 10654 | 11085 | 8440 | 6791 | 6432 | 9612 | 7044 | 9340 | 6224 | 7459 | 8600 |
| | | R² | **0.76** | **0.91** | **0.83** | **0.76** | **0.86** | **0.89** | **0.88** | **0.56** | **0.70** | **0.86** | **0.57** | **0.89** | **0.55** | **0.91** | **0.89** |
| | | RMSE | 10.78 | 7.96 | 10.60 | 11.14 | 9.78 | 8.40 | 9.65 | 9.29 | 6.79 | 9.43 | 7.51 | 8.33 | 7.74 | 8.73 | 8.11 |
| | | Slop | 1.47 | 1.32 | 1.22 | 1.27 | 1.24 | 1.25 | 1.39 | 1.29 | 1.26 | 1.31 | 0.96 | 1.27 | 1.16 | 1.33 | 1.24 |
| | | Intercept | -5.25 | -1.97 | -1.77 | -2.47 | -2.26 | -2.77 | -2.69 | -2.26 | -1.40 | -2.13 | 0.19 | -2.60 | -1.41 | -1.32 | -2.09 |

| **F. Ogden- South Ogden** | | | PurpleAir sensor ID | | | | | | |
|---|---|---|---|---|---|---|---|---|---|
| | | | 465 | 1104 | 5178 | 5454 | 6604 | 7858 | 7860 |
| EPA AQMS ID | 49-57-2-5 | Distance (km) | 4.15 | 3.95 | 3.92 | 3.26 | 3.95 | 2.72 | 3.15 |
| | | Obs (h) | 5127 | 7679 | 6944 | 5105 | 5662 | 6106 | 6219 |
| | | R² | 0.11 | 0.36 | 0.34 | 0.16 | 0.30 | 0.36 | 0.36 |
| | | RMSE | 9.08 | 9.15 | 9.27 | 8.27 | 10.51 | 10.60 | 9.68 |
| | | Slop | 0.21 | 0.68 | 0.64 | 0.36 | 0.62 | 0.73 | 0.66 |
| | | Intercept | 4.65 | 2.27 | 3.80 | 3.54 | 2.71 | 2.86 | 2.68 |

| **G. Lindon - Orem** | | | PurpleAir sensor ID | | | | | | | | | | | |
|---|---|---|---|---|---|---|---|---|---|---|---|---|---|---|
| | | | 5135 | 5143 | 5145 | 5728 | 5732 | 5736 | 5750 | 5754 | 5760 | 6304 | 6948 | 6986 |
| EPA AQMS ID | 49-49-4001-5 | Distance (km) | 1.91 | 4.93 | 3.75 | 3.36 | 3.92 | 4.81 | 4.43 | 4.12 | 3.48 | 4.27 | 1.74 | 0.4 |
| | | Obs (h) | 8388 | 3911 | 3465 | 7242 | 7408 | 7283 | 6224 | 6060 | 1626 | 7850 | 2963 | 4925 |
| | | R² | 0.22 | **0.50** | 0.20 | 0.49 | **0.50** | 0.43 | **0.55** | **0.51** | **0.66** | 0.48 | **0.58** | **0.52** |
| | | RMSE | 0.41 | 4.04 | 4.43 | 8.97 | 9.16 | 8.56 | 7.86 | 6.72 | 8.60 | 7.86 | 4.61 | 8.97 |
| | | Slop | 0.03 | 0.75 | 0.71 | 1.19 | 1.27 | 1.11 | 1.15 | 0.95 | 3.12 | 1.05 | 0.59 | 1.25 |



| | | Intercept | 0.09 | -0.41 | 0.77 | 0.54 | 0.92 | 0.75 | 0.13 | -0.01 | -3.54 | 0.31 | 0.59 | 1.09 |

| H. Salt Lake City | | | PurpleAir sensor ID | | | | | | | | | | | | |
|---|---|---|---|---|---|---|---|---|---|---|---|---|---|---|---|
| | | | 884 | 3388 | 5014 | 5460 | 5742 | 5802 | 5990 | 6078 | 6356 | 6360 | 6434 | 6608 | 6622 | 10050 |
| EPA AQMS ID | 49-35-3006-4 | Distance (km) | 4.95 | 4.29 | 4.47 | 1.34 | 2.21 | 4.49 | 4.78 | 3.20 | 4.33 | 3.42 | 4.48 | 4.58 | 0.87 | 3.67 |
| | | Obs (hours) | 13524 | 9283 | 8570 | 7450 | 6074 | 2126 | 3926 | 1200 | 7766 | 7766 | 7541 | 6944 | 6241 | 5614 |
| | | $R^2$ | **0.72** | **0.72** | **0.78** | **0.81** | **0.72** | 0.37 | **0.70** | 0.40 | **0.77** | **0.77** | **0.73** | **0.63** | **0.80** | **0.77** |
| | | RMSE | 6.14 | 6.45 | 6.85 | 5.00 | 6.37 | 3.89 | 7.81 | 4.10 | 5.70 | 5.39 | 7.32 | 7.17 | 5.43 | 6.94 |
| | | Slop | 1.36 | 1.24 | 1.51 | 1.41 | 1.31 | 0.79 | 1.34 | 0.78 | 1.40 | 1.33 | 1.58 | 1.22 | 1.43 | 1.58 |
| | | Intercept | -2.45 | -2.21 | -2.18 | -3.06 | -2.15 | -0.41 | -1.87 | -0.38 | -3.13 | -2.75 | -1.73 | -2.62 | -2.94 | -1.74 |
| | 49-35-3006-5 | Distance (km) | 4.95 | 4.29 | 4.47 | 1.34 | 2.21 | 4.49 | 4.78 | 3.20 | 4.33 | 3.42 | 4.48 | 4.58 | 0.87 | 3.67 |
| | | Obs (h) | 13975 | 10158 | 9431 | 8022 | 6748 | 2570 | 3981 | 1224 | 7982 | 8037 | 7808 | 7142 | 6421 | 5736 |
| | | $R^2$ | **0.68** | **0.67** | **0.72** | **0.70** | **0.64** | 0.37 | **0.65** | 0.20 | **0.67** | **0.68** | **0.66** | **0.55** | **0.72** | **0.71** |
| | | RMSE | 6.83 | 7.11 | 7.98 | 6.09 | 7.01 | 5.18 | 8.47 | 4.72 | 6.77 | 6.31 | 8.00 | 7.76 | 6.20 | 7.74 |
| | | Slop | 1.31 | 1.18 | 1.46 | 1.39 | 1.30 | 1.06 | 1.35 | 0.50 | 1.37 | 1.31 | 1.58 | 1.17 | 1.42 | 1.58 |
| | | Intercept | -1.10 | -0.86 | -0.45 | -1.69 | -1.01 | -0.67 | -0.50 | 1.39 | -1.77 | -1.51 | -0.45 | -1.29 | -1.80 | -0.20 |



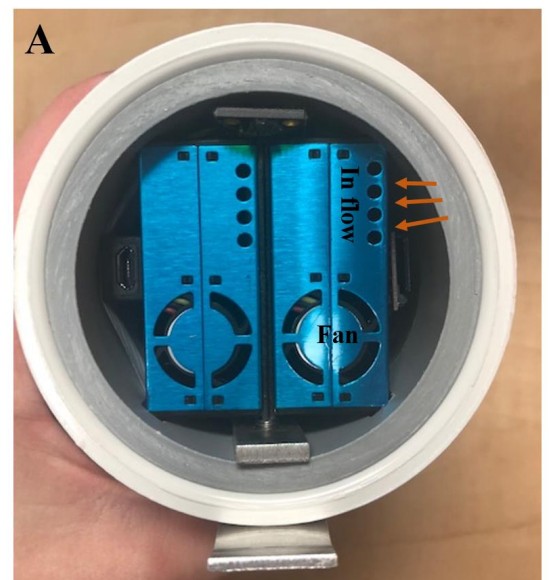

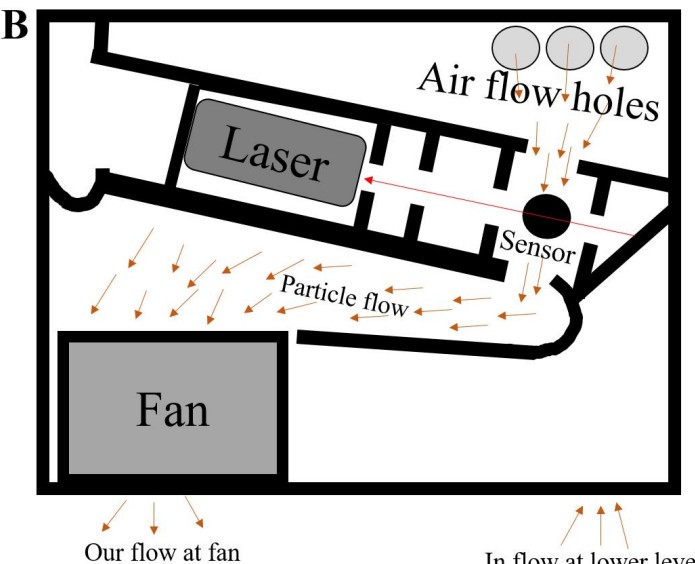





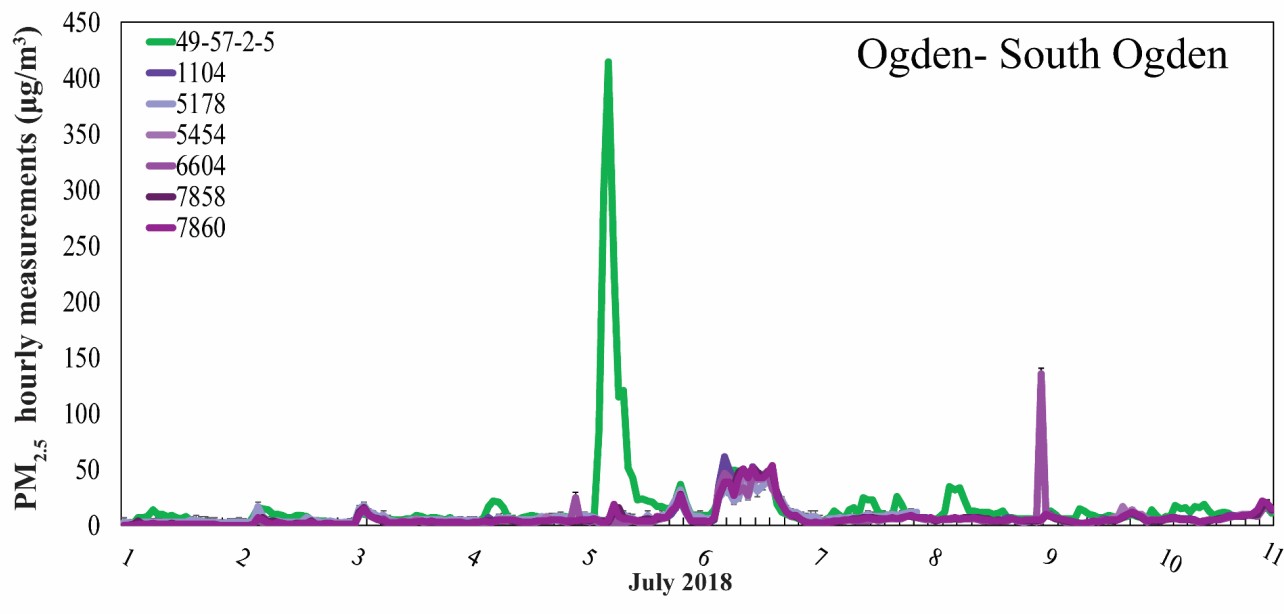





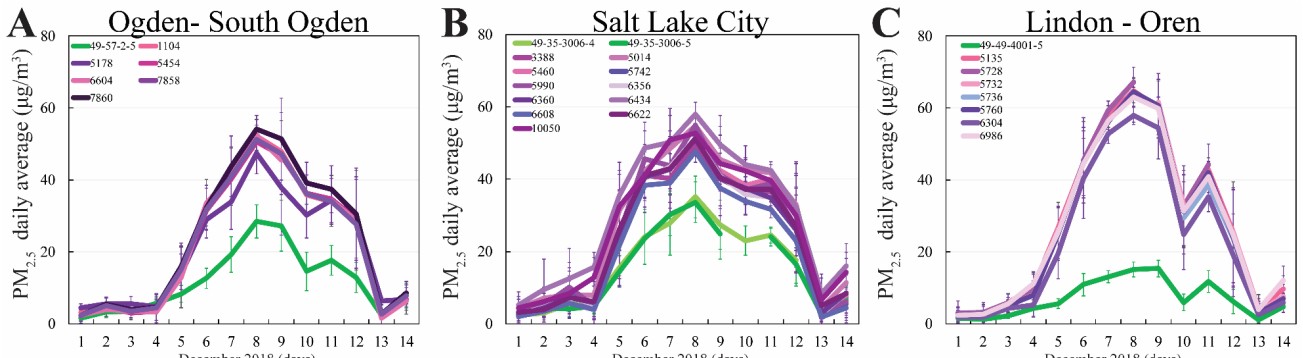




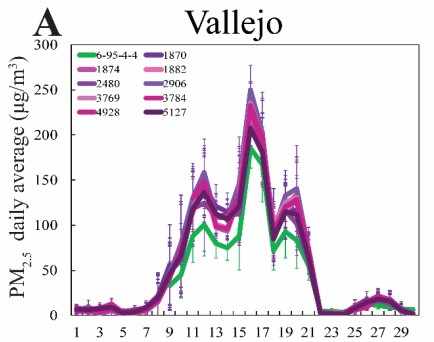

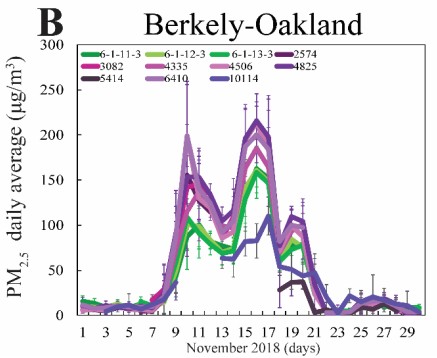

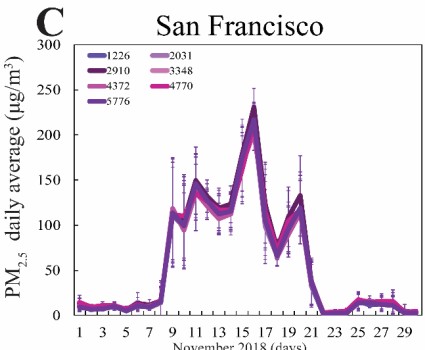





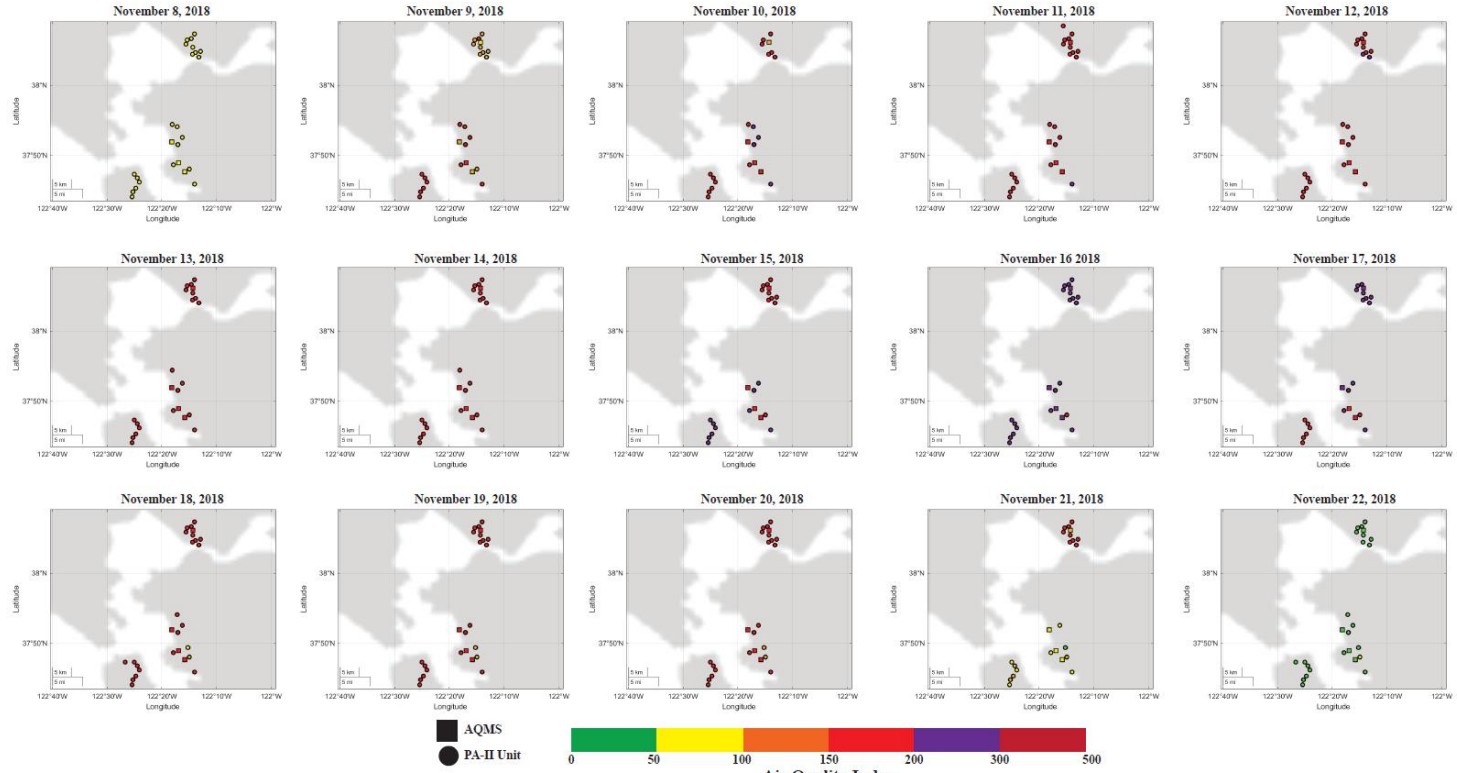