# Peer review of "Measurements of PM2.5 with PurpleAir under atmospheric conditions"

_Atmospheric Measurement Techniques, 2019_

## Referee Comment (RC1) · Anonymous Referee #1 · 10 Dec 2019

The work by Ardon-Dryer at al present a large database of PM2.5 mass concentrations collected by a low-cost sensor, the Purple Air PA-II unit across eight locations in the USA. At each location, there were measurements from multiple PA-II units, and the closest air quality monitoring station (AQMS) data was also acquired. The authors have performed a large amount of analysis on this dataset comparing the PA-II to reference instrumentation. However, it was not clear to me what the scientific novelty of the paper was, as there have been a number of papers already that evaluated the Purple Air sensor, as mentioned in the introduction. The authors state that the aim was to 'examine how PA-II units perform under atmospheric conditions when exposed to a variety of pollutants and PM2.5 concentrations', yet this is a rather vague aim, that this dataset may not be suitable to answer. This is a great dataset that could be

used to investigate a number of interesting questions regarding low-cost sensors and their calibration and suitability for large scale deployment. I feel that this paper suffers from a lack of focus and could be improved if the authors articulated and addressed more novel, detailed and specific aims and objectives. This leads to another area that could be improved, as most of the analysis is rather descriptive and lacking in depth. In my opinion, it is not enough to just present the regression analysis for all the PA-II units (i.e. r2, slopes etc) to the AQMS instruments. For example, there could have been more analysis on why there was a large range in observed r2 between all the unit and the AQMS? Was there any common factors for units that had a poor or good correlation with the AQMS? Did the actual reference instrument at the AQMS site affect the correlation (e.g. between FRM and FEM instruments)? I would have also like to have seen more focus on the observed slope between PA-II and the AQMS, as this is a better indicator of the accuracy of the PA-II than the correlation co-efficient (r2).

One of the key issues with this dataset, as acknowledged by the authors in section 3.3.3, was that the PA-II units were not co-located with each other or the AQMS and could therefore diminish the ability to compare the PA-II to reference instruments. Unfortunately, in my opinion the authors did not adequately address this issue. It would have been interesting if a more in-depth analysis of how the PA-II relationship with reference instrument varied as function of distance, as this would be of great interest to the community. The paper is well written and clearly presented but the large volume of data presented did make it difficult to follow at times. For example, the tables are too big, and could do with either being separated by city, or only the pertinent information being included.

In addition to the above, number of more detailed comments are given below Abstract: When you state that the units had good agreement it is important to back this up with numbers, such as giving the slopes, r2 etc. This generally true throughout the paper.

Page 2, line 63: In addition, the authors could reference Crilley et al 2018 and Di Antonio et al 2018 for possible solutions to the RH effect on low-cost PM sensors.

Page 5, line 152: this paragraph could instead be presented as a table. Furthermore, it may also help the reader if you were to give the AQMS and PA-11 units more accessible names. For example, the Pittsburgh AQMS could be P-AQ-1 and 2, and the PA-II units, P-PA-1, 2, 3 etc

Section 2.4: more info is needed on the data analysis, what sort of regression analysis did you do? In what computer program? Which AQMS station did you use, the closest or each one for a given city?

Section 2.6: I do not see the point of calculating the AQI when the point of this article is to compare the measurements between the PA-II and reference instruments. If they report the same concentration, wont they give the same AQI? I think you should just focus on reported concentrations.

Section 3.1.1: If Fig 2 is on page 29, then this is not a distribution but a time series of reported concentrations. A distribution to me implies a histogram, please correct the naming. Also why did the AQMS report higher PM2.5 concentrations at Berkley, Ogden, Linden and Salt Lake City compared to all the PA-II units during the first half of 2018? Understanding why the relationship changed is important for knowing the parameters that affect the PA-II measurements.

Page 8, line 236: the authors state "These high correlation values and relatively low RMSE indicate that although the PA-II units and the AQMS are not co-located, they still tend to behave in a similar way." Why do think this was the case?

Page 8, line 242: I do not understand what you mean by instrument efficiency?

Page 8, line 250: why did you subset the data below 40 ug m-3?

Page 12, line 364. In the previous paragraph you state that RH is a more important parameter than temperature when considering potential artefacts for the PA-II, so why compare to temperature?

Page 12. Line 381: I do not agree with this statement as you have not able to test the

precision of the PA-II as they were not co-located. The precision of the PA-II units would be tested by how well each PA unit agree with each other at a given RH, but you have looked for correlation between RH and PA-II reported PM2.5. this does not indicate the precision of the PA-II only if there was a relationship between RH and reported PM2.5 concentrations.

Page 13, line 418: where the slopes between the PA-II and AQMS instruments affected by distance?

Section 3.4: I think that section could be improved by including some recommendations based on your findings from this study.

Page 14, line 433: please call it instrument drift, as instrument efficiency is meaningless in this context.

---

## Referee Comment (RC2) · Anonymous Referee #2 · 26 Mar 2020

This is an informative manuscript that evaluates the performance of networks of the PurpleAir PA-II low-cost aerosol sensor in real-world use. These sensors are commonly purchased by private citizens and installed, sometimes haphazardly, in residential and commercial neighborhoods. They are quite low-cost (<$300/unit) and data from these sensors could be used to increase understanding of the spatial distribution of PM2.5 and supplement more comprehensive, but much more costly and less ubiquitous, air quality monitoring stations (AQMS). The real question is whether these sensors provide data of adequate quality to be useful.

The paper is generally clear and well-written, and it makes a strong case that the sensors have value and can provide scientifically useful information, at least under the conditions evaluated. It is also nice to see a high school student involved in the study.

[Figure]

That said, there are some changes that need to be made to improve the manuscript. In particular, the evaluation of the sensitivity of the sensors to relative humidity (RH) and temperature (T) needs to be reworked, and some of the information in the tables could be presented more effectively with graphics.

Below are major concerns, followed by a couple of minor issues. I have not checked the references for completeness.

1) In section 3.3.1, the effect of RH and T on unit performance are evaluated by regressing these values against the PM2.5 values from the PA-II units. Unsurprisingly, there was no significant correlation against either of these parameters. Instead, what needs to be compared is RH and T against the *difference* between the PA-II units and the nearest AQMS values. Biases associated with T and RH are minimized in the AQMS sensors but would show up in the PA-II sensors, which do not control sample RH or T (although T is higher inside the sensing elements; thus we would expect RH to be reduced significantly below ambient). Any large bias associated with RH or ambient T should show up in this comparison (except see minor comment (b) below).

2) There are a lot of values in tables in this manuscript, many of which really belong in the supplemental information. I would much prefer to see a new figure with scatterplots of each sensor against the AQMS values in the main text, and move Figs. S1 amd S3 there as well. The detail in the tables should be moved to the SI.

3) The linear regressions should be performed with "2-sided" regressions because there is uncertainty in both the x and y values of the scatterplots. Standard linear regression assumes uncertainty only in the y values. I also suggest you remove obvious outliers (for example, the July 4th fireworks smoke) before performing regressions; these outliers can severely torque the slopes and r2 values.

4) There is lack of specificity in the abstract and throughout the text about "co-located" and "same location". I was quite confused when first reading the abstract, because it says that this manuscript reports analysis of PA-II units that are not "co-located" with

AQMS sites, but then in the next sentence that "we selected eight different locations, where each location contains multiple PA-II units (minimum of seven per location, a total of 86 units) and at least one AQMS (total of 14)." This sounds to me like "co-located" because you have not specified the criteria used for selecting PA-II units. I suggest you use "nearby" or "regional" rather than "location" throughout the text to avoid confusion. And please define the distance criteria for which PA-II units were selected for comparison with AQMS instruments.

5) You may want to explore the seasonality of differences between the PA-II units and the AQMS values. For example, in winter in Utah, I would expect big gradients between airport sensors on the flat plains and residential sensors on the slopes. This may become evident in the analysis I suggest in comment (1) above.

The take-home message to me is that the sensors are surprisingly good. If outliers are removed and sensors compared against others in the region for basic quality control, there is scientific value in the spatial information gained from networks of these cheap sensors.

Minor comments:

a) Lines 151-164. These are not needed; this information is already in the tables.

b) In Sect. 3.2.2., these differences between the AQMS values and the PA-II data in Utah in winter may be associated with the volatility of ammonium nitrate, which dominates the aerosol composition there (Womack et al., https://doi.org/10.1029/2019GL082028). The PA-II instrument would be less likely to volatilize ammonium nitrate, while the NAAQS FRM does volatilize it (Grover et al., https://doi.org/10.1029/2004JD004995).

---

## Author Comment (AC1) · 10 Jul 2020

We would like to thank anonymous reviewers 2 for the helpful comments and suggestions. In line with the reviewer comments and suggestions, and in line with new publications that were published while this paper was under review, we modify and revised the manuscript. Below are all the comments (in bold) followed by the replies. The parts that are in italic are corrections that are included in the revised version of the paper.

Sincerely,

Karin Ardon-Dryer

**Response to Reviewer 2**

**This is an informative manuscript that evaluates the performance of networks of the PurpleAir PA-II low-cost aerosol sensor in real-world use. These sensors are commonly purchased by private citizens and installed, sometimes haphazardly, in residential and commercial neighborhoods. They are quite low-cost (<\$300/unit) and data from these sensors could be used to increase understanding of the spatial distribution of PM2.5 and supplement more comprehensive, but much more costly and less ubiquitous, air quality monitoring stations (AQMS). The real question is whether these sensors provide data of adequate quality to be useful. The paper is generally clear and well-written, and it makes a strong case that the sensors have value and can provide scientifically useful information, at least under the conditions evaluated. It is also nice to see a high school student involved in the study. That said, there are some changes that need to be made to improve the manuscript. In particular, the evaluation of the sensitivity of the sensors to relative humidity (RH) and temperature (T) needs to be reworked, and some of the information in the tables could be presented more effectively with graphics. Below are major concerns, followed by a couple of minor issues. I have not checked the references for completeness.**

**1) In section 3.3.1, the effect of RH and T on unit performance are evaluated by regressing these values against the PM2.5 values from the PA-II units. Unsurprisingly, there was no significant correlation against either of these parameters. Instead, what needs to be compared is RH and T against the \*difference\* between the PA-II units and the nearest AQMS values. Biases associated with T and RH are minimized in the AQMS sensors but**

**would show up in the PA-II sensors, which do not control sample RH or T (although T is higher inside the sensing elements; thus we would expect RH to be reduced significantly below ambient). Any large bias associated with RH or ambient T should show up in this comparison (except see minor comment (b) below).**

We took into consideration the reviewer comments, therefore we made extensive changes in our manuscript. First, we added an evaluation of the PA-II sensitivity to RH and T, for co-located PA-II units with AQMS (Fig S4). We also added an entire paragraph that discusses the impact of RH and T on the PA-II. Also, based on the reviewer's comments and suggestions as well as new publications that were published while the original manuscript was under review, we added an entirely new analysis to the paper. We performed a multivariate linear regression (MLR) on the co-located units (PA-II and AQMS, that were at a distance up to 1.1 km) and used the coefficient from the MLR to correct that additional PA-II unit measurements taken in the same region. This correction of the PA-II PM2.5 values improve the comparison between the PA-II units and the AQMS as well as between the PA-II to other PA-II units, as showed by improving the slop and reduction of the root mean square error (RMSE) and mean absolute error (MAE) values.

The following information was added to the manuscript

*The overestimating raises questions about the accuracy of the PA-II units. According to PurpleAir (PurpleAir, personal communication, 2019) the company does not calibrate the PA-II units; instead, before each PA-II unit is sent out to a customer, the company performs a comparison test with a dozen PA-II units to find and remove outliers from the shipment (PurpleAir, personal communication, 2019). Previous studies suggested that part of the problem with the PA-II unit results from the optical particle counter being impacted by changes of RH (Crilley et al., 2018; Malings et al., 2020; Magi et al., 2020). Water vapor can condense on aerosol particles, making them grow hygroscopically under high RH conditions (Lundgren and Cooper, 1969). The PA-II units do not have any heater or dryer at their inlets to remove water from the sample before measuring the particles; therefore, deliquescent or hygroscopic growth of particles, mainly under high RH conditions, can lead to higher reported PM concentrations (Di Antonio, 2018; Jayaratne et al., 2018; Bi et al., 2020), which ends as an overestimate of the PM compared to the reference units. Weather conditions can impact the values reported by low-cost sensors (Morawska et al.,*

2018). Changes in T or RH have been found to affect the performance of the PA-II units, especially under atmospheric conditions, as they cannot be controlled (Bi et al., 2020). Therefore, MLR between a PA-II, and an AQMS, which also considers changes of T and RH, can help correct the reported PM$_{2.5}$ values of the co-located PA-II units. Similar corrections have been suggested and implemented in other locations with PA-II units (Bi et al., 2020; Magi et al., 2020) and other low-cost sensors (Malings et al., 2020). Most of these studies focus on co-located units or on units that were up to 1 km from the reference unit.

Calculations of the ratio between the measured PM$_{2.5}$ from the PA-II to the AQMS as a function of T and RH, known as a hunidogram, were performed (Fig S4). Some of the PA-II units seem to be impacted by T and RH more than others; these units also had relatively low $R^2$ values with the AQMS unit, as in the case of DE-PA-6 in Denver (Fig. S4A).

[Figure]

**Figure S4:** *Ratio between measured PM$_{2.5}$ from PA-II to the AQMS, as a function of temperature and relative humidity (hunidogram) for all collocated PA-II and AQMS pairs. Information on the distance and $R^2$ values between the two presented in each plot.*

**2) There are a lot of values in tables in this manuscript, many of which really belong in the supplemental information. I would much prefer to see a new figure with scatterplots of each sensor against the AQMS values in the main text, and move Figs. S1 amd S3 there as well. The detail in the tables should be moved to the SI.**

Per the reviewer's suggestions all tables were moved to the supplement (Now Tables S1-S3) and scatterplot of the PA-II compared to the AQMS were added to the manuscript (Fig. 4 and Fig.5). We compared the co-located units before and after we performed the MLR (Fig 4,) and observed the difference before and after we applied the coefficients from the MLR to the rest of the PA-II units (Fig 5.)

As suggested by the reviewer the figure with the map (originally Fig S1) was moved to the main manuscript, and it is now Fig. 2. Figure S3 was also moved to the main manuscript, it is now Fig. 7. We made changes in the figure, we evaluated the impact of the distance on the $R^2$, RMSE, MAE, and the slope values. This was performed both between the PA-II to the nearest AQMS as well as between the PA-II units.

[Figure]

***Figure 7***: *Comparison of distance (km) between PA-II to its nearest AQMS in all regions (A) and between each PA-II unit to all other PA-II units per region (B) to $R^2$, RMSE, MAE and slope values received from the $PM_{2.5}$ hourly measurements comparison.*

**3) The linear regressions should be performed with "2-sided" regressions because there is uncertainty in both the x and y values of the scatterplots. Standard linear regression assumes**

**uncertainty only in the y values. I also suggest you remove obvious outliers (for example, the July 4th fireworks smoke) before performing regressions; these outliers can severely torque the slopes and r2 values.**

We apologize but we were unsure about the reviewer meaning for 2-sided regression and why he considers the AQMS measurements as uncertain. Our study, like others treats the AQMS as an absolute and does not question the validity or accuracy of its measurements.

As suggested by the reviewer we removed all the outlier's events before the statistical tests, we also performed an analysis that allows us to remove outlier PA-II units. A new section was added to the manuscripts about describing both.

The following information was added to the manuscript

*2.5. Remove of outlier PA-II units and irregular hours*

*The first step was to identify outliers among the PA-II units, per region, meaning PA-II units that behave differently from the other PA-II units in their region. By comparing $R^2$ between the $PM_{2.5}$ values measured by each pair of PA-II units, using a linear regression, we identified the outlier units. A PA-II unit that did not have an $R^2 \geq 0.75$ with at least 75% of the other PA-II units in its region was considered an outlier unit, and therefore was removed from future analysis (Fig. S1 shows a comparison for each of the four regions). Only one unit from SF (SF-PA-9, see Fig. S1B) had very low $R^2$ when compared to all other PA-II units. Most PA-II units had high $R^2$ values (>0.9) with the other units. Irregular $PM_{2.5}$ hourly measurements were removed from all units (PA-II and AQMS). These irregular hourly measurements were identified as a large single hourly increase of $PM_{2.5}$ values (>70 μg m$^{-3}$) that was not measured by any other unit in the region. Such a large increase was caused most likely by a local source near a specific unit, such as a small-scale fire, lawn mower, barbecue, cigarette smoke, or fireworks (Zheng et al., 2018), and attributed to the location of many of the PA-II units in a residential area. Firework events were removed, as they were very localized events and were measured by a single unit. Overall, less than 0.03% of the hourly $PM_{2.5}$ measurements identified as irregular hours were removed from different PA-II and AQMS units.*

**4) There is lack of specificity in the abstract and throughout the text about "co-located" and "same location". I was quite confused when first reading the abstract, because it says that this manuscript reports analysis of PA-II units that are not "co-located" with AQMS sites, but then in the next sentence that "we selected eight different locations, where each location contains multiple PA-II units (minimum of seven per location, a total of 86 units) and at least one AQMS (total of 14)." This sounds to me like "colocated" because you have not specified the criteria used for selecting PA-II units. I suggest you use "nearby" or "regional" rather than "location" throughout the text to avoid confusion. And please define the distance criteria for which PA-II units were selected for comparison with AQMS instruments.**

We apologize that our lack of clarity about the location aspect of the units. As the reviewer suggested we added more clarification to the manuscript. Co-located units are PA-II and AQMS units that are up to 1.1 km between each other, this is similar range to what was done by Bi et al. (2020) . We also changed the use of the word location to region as suggested by the reviewer. In addition, we provide the extract criteria for a distance that was used in our analysis to define each region.

This information was added to the abstract:
*For this study, we selected four different regions, each containing multiple PA-II units (minimum of seven per region). In addition, each region needed to have at least one AQMS unit that was co-located with at least one PA-II unit, all units needed to be at a distance of up to 5 km from an AQMS unit and have up to 10 km between each other.*

**5) You may want to explore the seasonality of differences between the PA-II units and the AQMS values. For example, in winter in Utah, I would expect big gradients between airport sensors on the flat plains and residential sensors on the slopes. This may become evident in the analysis I suggest in comment (1) above.**

Per the reviewer comments we analyzed the seasonality differences between the PA-II units and the AQMS values in all four regions, we attempted to identify the impact of T and RH as suggested by the reviewer. All regions had lower $R^2$, RMSE and MAE values in the spring compared to the

other seasons, however, this difference was not statistically significant for all cases. Next, we calculated the average RH and T for each season, and we compared it to the $R^2$, RMSE and MAE values. To our surprise there was no seasonal impact of RH or T on these values. We found that the lower $R^2$, RMSE and MAE values in the spring result from the overall lower PM2.5 values measured in that season for all four regions (as can be seen in Fig.3 in the manuscript). The PM concentrations had a stronger impact on the PA-II and AQMS comparisons than the T and RH had, therefore, we decided not to include this analysis in the manuscript.

As for the reviewer's example, we explored the spatial changes between the PA-II units, mainly in Salt Lack City, Utah as suggested by the reviewer. All the units that we used in the study were in residential area and not next to the airport. Overall, most sensors behaved in a similar way, as shown by the figure below. A similar range (bins of 5 µg m-3) of PM$_{2.5}$ concentration were measured by all the units. However, in the very few cases in which we observed some spatial differences (mainly in August 2018, as shown in the Figure below), we could determine the causes of these differences.

[Figure]

Fig 2. Time series of daily PM$_{2.5}$ measurements from the AQMS and PA-II units in Salt Lake City during 2018 (top). Measurements from AQMS are represented by the green lines and the PA-II units are indicated by purple lines, RH values represented by the gray dotted line. Maps of different days during 2018 with the spatial distribution of the daily PM$_{2.5}$ measurements (lower panel).

AQMS represented by the square and PA-II by round shape. Each color represents PM$_{2.5}$ values in bins of 5 μg m$^{-3}$.

**Minor comments:**

**a) Lines 151-164. These are not needed; this information is already in the tables.**

This entire paragraph was removed from the manuscript

**b) In Sect. 3.2.2., these differences between the AQMS values and the PA-II data in Utah in winter may be associated with the volatility of ammonium nitrate, which dominates the aerosol composition there (Womack et al., https://doi.org/10.1029/2019GL082028). The PA-II instrument would be less likely to volatilize ammonium nitrate, while the NAAQS FRM does volatilize it (Grover et al., https://doi.org/10.1029/2004JD004995).**

We would like to thank the reviewer for bringing up this point. The reviewer comments regarding the volatility of ammonium nitrate helped us to understand one of the causes for the increase of PM$_{2.5}$ in Salt Lake City during the winter months. We added this information to the manuscript

The following information was added to the manuscript

[revised manuscript text omitted]

---

## Author Comment (AC2) · 10 Jul 2020

We would like to thank anonymous reviewers 1 for the helpful comments and suggestions. In line with the reviewer comments and suggestions, and in line with new publications that were published while this paper was under review, we modify and revised the manuscript. Below are all the comments (in bold) followed by the replies. The parts that are in italic are corrections that are included in the revised version of the paper.

Sincerely,

Karin Ardon-Dryer

Response to Reviewer 1

**The work by Ardon-Dryer at al present a large database of PM2.5 mass concentrations collected by a low-cost sensor, the Purple Air PA-II unit across eight locations in the USA. At each location, there were measurements from multiple PA-II units, and the closest air quality monitoring station (AQMS) data was also acquired. The authors have performed a large amount of analysis on this dataset comparing the PA-II to reference instrumentation. However, it was not clear to me what the scientific novelty of the paper was, as there have been a number of papers already that evaluated the Purple Air sensor, as mentioned in the introduction. The authors state that the aim was to 'examine how PA-II units perform under atmospheric conditions when exposed to a variety of pollutants and PM2.5 concentrations', yet this is a rather vague aim, that this dataset may not be suitable to answer. This is a great dataset that could be used to investigate a number of interesting questions regarding low-cost sensors and their calibration and suitability for large scale deployment. I feel that this paper suffers from a lack of focus and could be improved if the authors articulated and addressed more novel, detailed and specific aims and objectives. This leads to another area that could be improved, as most of the analysis is rather descriptive and lacking in depth. In my opinion, it is not enough to just present the regression analysis for all the PA-II units (i.e. r2, slopes etc) to the AQMS instruments. For example, there could have been more analysis on why there was a large range in observed r2 between all the unit and the AQMS? Was there any common factors for units that had a poor or good correlation with the AQMS? Did the actual reference instrument at the AQMS site affect the correlation (e.g. between FRM and FEM instruments)? I would have also like to have seen more focus on the observed slope**

**between PA-II and the AQMS, as this is a better indicator of the accuracy of the PA-II than the correlation co-efficient (r2).**

We appreciate the reviewer's comment, based on the review comment we clarify our aims in the manuscripts.

The following information was added to the manuscript

*This study aims to examine how each PA-II unit performs under atmospheric conditions when exposed to a variety of pollutants and PM$_{2.5}$ concentrations (PM with an aerodynamic diameter smaller than 2.5 μm), when at a distance from the reference sensor. We examine how PA-II units perform in comparison to other PA-II units and Environmental Protection Agency (EPA) Air Quality Monitoring Stations (AQMSs) that are not co-located with them.*

*This study aims to examine how PA-II units perform under atmospheric conditions when exposed to a variety of pollutants and PM$_{2.5}$ concentrations. For the scope of this study, we chose to focus only on regions that contain at least one pair of co-located PA-II and AQMS units. Corrections of PM$_{2.5}$ values for co-located PA-II and AQMS units, based on MLR, were performed and applied to all the other PA-II units in that region. Comparison of PM$_{2.5}$ measurements taken by all units in each region, AQMSs and PA-II units (when PM$_{2.5}$  values were measured or corrected) are presented. The presented comparisons were done for both the entire study period and for specific events that we wanted to examine in greater detail.*

Regarding the comments about factors for units that had a poor or good correlation with the AQMS, In our original manuscript most of the PA-II units that had a low correlation with the AQMS units also suffer from low correlation with the other PA-II units, we decided to remove these units as we believe they are outliers. In the current dataset, there are only two units that had a lower correlation with the AQMSs, but these two units were borderline for our PA-II outlier test. Without these two units, most of the $R^2$ values will be >0.6. The evaluation test between the PA-II units will help identify PA-II units that are not performing well. A reduction in performance can occur over time or due to exposure to events with high PM, as described in Sayahi et al. (2019). This information was added to the manuscript.

The following information was added to the manuscript

*Overall, almost all the PA-II units had high correlation values when compared with the other PA-IIs or AQMSs in their region. Two PA-II units, SL-PA-6 and SL-PA-8 had low $R^2$ values with the AQMS, they also had a relatively low correlation with the other PA-II units. It is feasible, that if stricter rules for identifying outlier PA-II units were in use, these two units would have been considered as such and subsequently removed from the data set.*

As for the comment **Did the actual reference instrument at the AQMS site affect the correlation (e.g. between FRM and FEM instruments).**
All the AQMSs that were used in this work were of FEM type, their selection was based on the distance that was used in previous works (e.g. Bi et al., 2020). Therefore, we could not evaluate that difference (FEM vs FRM).

Regarding the reviewer comments *that like to have seen more focus on the observed slope between PA-II and the AQMS, as this is a better indicator of the accuracy of the PA-II than the correlation co-efficient (r2).* Information on the slope was added to the manuscripts for all comparisons as shown in Table 1, Table S3, and Fig. 7.

**One of the key issues with this dataset, as acknowledged by the authors in section 3.3.3, was that the PA-II units were not co-located with each other or the AQMS and could therefore diminish the ability to compare the PA-II to reference instruments. Unfortunately, in my opinion the authors did not adequately address this issue. It would have been interesting if a more in-depth analysis of how the PA-II relationship with reference instrument varied as function of distance, as this would be of great interest to the community.**

The goal of this paper was to observe $PM_{2.5}$ measurements using the PA-II units, these units installed by citizens are for the most found in residential locations across the United States, therefore only a handful are co-located with an AQMS, and in fact many of the regions in which PA-IIs are deployed do not have even a single reference unit. Previous works have examined the efficiency of the PA-II unit by comparing it to a co-located AQMS or in laboratory conditions. For

the purpose of this study we defined a co-located pair as a PA-II that is up to 1.1 km away from an AQMS, the selection of this distance is based on the work of Bi et al. (2020). A major addition to this revision is the implementation of a data correction process that was applied to the PA-II measurements. This correction process was well documents by Bi et al. (2020) and Magi et al. (2020) for both PA-II units and by Malings et al. (2020)for other low cost sensors.

As for the impact of the distance of the units, we did not find that the distance between the units impacted the behavior and comparison of the unit, yet we only evaluate a distance of up to 5 km from an AQMS and up to 10 km between PA-IIs, units which will be far away may have a different impact, but evaluating that would be beyond the scope of this work.

**The paper is well written and clearly presented but the large volume of data presented did make it difficult to follow at times. For example, the tables are too big, and could do with either being separated by city, or only the pertinent information being included.**

We modified the provided tables and information in the text. Each table now represents a single region and does not include more than two parameters.

**In addition to the above, number of more detailed comments are given below Abstract: When you state that the units had good agreement it is important to back this up with numbers, such as giving the slopes, r2 etc. This generally true throughout the paper.**

Information on the comparisons between units was added to the manuscript per region. Values of the R-squared ($R^2$), root mean square error (RMSE), mean absolute error (MAE) as well as the best fit information, including the slope, are provided in the revised manscript.

Based on the reviewer comments the following information was added to the manuscript
*In most cases, the AQMSs and the PA-II units were found to be in good agreement (75% of the comparisons had a $R^2 > 0.8$)*

**Page 2, line 63: In addition, the authors could reference Crilley et al 2018 and Di Antonio et al 2018 for possible solutions to the RH effect on low-cost PM sensors.**

Both references were added to the manuscript

*Previous studies suggested that part of the problem with the PA-II unit results from the optical particle counter being impacted by changes of RH (Crilley et al., 2018; Malings et al., 2020; Magi et al., 2020). …. deliquescent or hygroscopic growth of particles, mainly under high RH conditions, can lead to higher reported PM concentrations (Di Antonio, 2018; Jayaratne et al., 2018; Bi et al., 2020), which ends as an overestimate of the PM compared to the reference units.*

**Page 5, line 152: this paragraph could instead be presented as a table. Furthermore, it may also help the reader if you were to give the AQMS and PA-11 units more accessible names. For example, the Pittsburgh AQMS could be P-AQ-1 and 2, and the PA-II units, P-PA-1, 2, 3 etc**

Per the reviewer's comments, the entire paragraph was removed from the manuscript. Also, all unit's names are now represented by location and instrument code as well as running ID number, as specified in Table S1.

*For simplifications, each region was defined by two letters to represent its name (DE for Denver, SF for San Francisco, VA for Vallejo, and SL for Salt Lake City). Also, each unit type received a two letter code (AQ for AQMS and PA for PA-II). Each unit received a number instead of an ID, as shown in Table S1.*

**Section 2.4: more info is needed on the data analysis, what sort of regression analysis did you do? In what computer program? Which AQMS station did you use, the closest or each one for a given city?**

In the manuscript, we describe that we used Multivariate linear regression (MLR) models between the $PM_{2.5}$ values of the co-located PA-II and AQMS with T and RH. In addition, all the analyses were performed using Matlab and Excel. This information was added to the manuscript

*To evaluate the similarities and differences between the PA-II units and the AQMSs and other PA-II units, a set of calculations and comparisons was performed using Matlab and Excel*

Regarding the AQMS database, we downloaded the entire data set of the hourly $PM_{2.5}$ for all AQMS units that were active during the study period. Using a distance calculation, we were able to identify regions with multiple PA-II units as well as at least one AQMS. We added this information to the manuscript

*Hourly measurements of $PM_{2.5}$ (FRM/FEM Mass code - 88101 file) from all AQMSs collected by the EPA from January 1, 2017, to December 31, 2018, were selected from the EPA website (https://aqs.epa.gov/api).*

*By using the JSON file for the PA-II units and the 88101 file for the AQMS, we calculated the distances between all the units to identify regions with multiple PA-II units (a minimum of five units) and at least one AQMS. At least one AQMS unit needed to be at a distance of 1.1 km from at least one PA-II unit (defined as a co-located pair, a similar range used by Bi et al., 2020). All the units in these regions needed to be active during the designated time period of January 1, 2017, to December 31, 2018. In each region PA-II units needed to be less than 5 km from at least one AQMS unit and up to 10 km from each other.*

*Four different regions containing a total of seven different AQMSs (all FEM type) and 46 different PA-II units were identified:*

**Section 2.6: I do not see the point of calculating the AQI when the point of this article is to compare the measurements between the PA-II and reference instruments. If they report the same concentration, wont they give the same AQI? I think you should just focus on reported concentrations.**

AQI information was removed from the manuscript per the reviewer's comment.

**Section 3.1.1: If Fig 2 is on page 29, then this is not a distribution but a time series of reported concentrations. A distribution to me implies a histogram, please correct the naming. Also why did the AQMS report higher PM2.5 concentrations at Berkley, Ogden, Linden and Salt Lake City compared to all the PA-II units during the first half of 2018? Understanding why the relationship changed is important for knowing the parameters that affect the PA-II measurements.**

This plot is now presented as a time series per the reviewer's comment.

*Time series of daily PM$_{2.5}$ values for each unit at each of the four regions are presented in Fig. 3.*

Some of the regions mentioned in the reviewer comment have been removed from the manuscript as they were missing a co-located AQMS. For the remaining regions, the higher AQMS measurements are attributed to what seems to be a connection with days that have low RH values resulting in lower PM$_{2.5}$ values being measured by the PA-II units. We also believe that chemical analysis during for these times could help understand the difference between the AQMS and PA-II, unfortunately, such analysis will be beyond the scope of this study.

*In some cases, the AQMS measured higher PM$_{2.5}$ daily values compared to the PA-II units, mainly at days with low PM$_{2.5}$ values, as seen in April - June 2018 in Vallejo (Fig. 3C) and Salt Lake City (Fig. 3D). These differences were observed mainly in days with low RH values (Fig. S3).*

[Figure]

***Figure S3:*** *Time series of daily PM$_{2.5}$ measurements from the AQMS and PA-II units in Vallejo (A), and Salt Lake City (B) during April-May 2018. Measurements from AQMS are represented by the green lines and the PA-II units are indicated by purple lines. Relative Humidity values represented by the gray dotted line.*

**Page 8, line 236: the authors state "These high correlation values and relatively low RMSE indicate that although the PA-II units and the AQMS are not co-located, they still tend to behave in a similar way." Why do think this was the case?**

This sentence was removed from the current manuscript.

**Page 8, line 242: I do not understand what you mean by instrument efficiency?**

This sentence was removed from the current manuscript.

**Page 8, line 250: why did you subset the data below 40 ug m-3?**

This analysis was removed from the current manuscript. Originally, we set 40 ug m-3 as the maximum point for the study as the work of Sayahi et al. (2019) suggests that above it the PA-II measurements are impacted by the high PM concentrations. Meaning at lower PM concentration we will find a better correlation between the PA-II and the AQMS.

**Page 12, line 364. In the previous paragraph you state that RH is a more important parameter than temperature when considering potential artefacts for the PA-II, so why compare to temperature?**

This sentence was removed from the current manuscript.

The original sentence was based on findings from several papers. We originally compared the temperature in order to prove our theory that temperature is not as important. However, during the time that our original manuscript was under review several new papers were published which in turn made us make extensive changes in our manuscript regarding the impact of RH and T. We added a hunidogram and a plot that investigate the impact of T in each of the co-located units (AQMS with PA-II, Fig S4). Some of the PA-II units might be impacted by both T and RH, this information was added to the manuscript

*Calculations of the ratio between the measured PM$_{2.5}$ from the PA-II to the AQMS as a function of T and RH, known as a hunidogram, were performed (Fig S4). Some of the PA-II units seem to be impacted by T and RH more than others; these units also had relatively low R$^2$ values with the AQMS unit, as in the case of DE-PA-6 in Denver (Fig. S4A).*

[Figure]

**Figure S4:** *Ratio between measured PM$_{2.5}$ from PA-II to the AQMS, as a function of temperature and relative humidity (hunidogram) for all collocated PA-II and AQMS pairs. Information on the distance and R$^2$ values between the two presented in each plot.*

We also added a multivariate linear regression (MLR). The MLR takes into account changes of T and RH. All the PA-II units' measurements were corrected based on the MLR of the co-located PA-II and AQMS, this information was added to the manuscript:

*Based on the MLR, the multivariable linear dependence of PA-II PM$_{2.5}$ on AQMS, RH and T created the predictors of PA-II as:*

$$PA - II(PM_{2.5}) = A_1 + A_2 AQMS(PM_{2.5}) + A_3 T + A_4 RH$$

*(1)*

*where $A_1$, $A_2$, $A_3$, and $A_4$ fit coefficients received from the MLR, PA-II ($PM_{2.5}$) and AQMS($PM_{2.5}$) are in units of $\mu g\ m^{-3}$, T is in Celsius, and RH is in percentage. Based on these parameters and fit coefficients, a calculation of the corrected PA-II $PM_{2.5}$ hourly values for each PA-II was performed using the following:*

$$PA - II(PM_{2.5}), corrected = \frac{PA - II(PM_{2.5}), uncorected - A_1 - A_3 T - A_4 RH}{A_2}$$

*(2)*

*Details of the coefficients received in the MLR as well as the regression output including $R^2$, RMSE, MAE, and slope for each correction of $PM_{2.5}$ values in the PA-II units, for each region, can be found Table 1. Figure 3 presents a comparison of the $PM_{2.5}$ values from the uncorrected PA-II unit to the AQMS as well as the PA-II $PM_{2.5}$ values hourly after correction, per region.*

**Page 12. Line 381: I do not agree with this statement as you have not able to test the precision of the PA-II as they were not co-located. The precision of the PA-II units would be tested by how well each PA unit agree with each other at a given RH, but you have looked for correlation between RH and PA-II reported PM2.5. this does not indicate the precision of the PA-II only if there was a relationship between RH and reported PM2.5 concentrations.**

This sentence was removed from the current manuscript.

**Page 13, line 418: where the slopes between the PA-II and AQMS instruments affected by distance?**

We did not find an impact of the distance on the slop., No impact was observed when the PA-II units were compared to the nearest AQMS in all regions, and no effect was found when comparing the PA-II to each other.

This information was added to the manuscript as text and figure:
*Because the AQMS and the PA-II units were not co-located, we wanted to verify whether the distance between all the units affected the $R^2$, RMSE, MAE and slope values. We compared the $R^{2,}$*

*RMSE, MAE and slope values received from the comparisons of hourly PM$_{2.5}$ measurements with the corresponding distances between the units (Fig. 7). There was no correlation between the two. Not when the PA-II units were compared to the nearest AQMS units (Fig. 7A), or between the PA-II units (Fig.7B), before or after the corrections of the PA-II PM$_{2.5}$ values. Therefore, the distance between the units did not impact the comparison.*

[Figure]

***Figure 7***: *Comparison of distance (km) between PA-II to its nearest AQMS in all regions (A) and between each PA-II unit to all other PA-II units per region (B) to R$^2$, RMSE, MAE and slope values received from the PM$_{2.5}$ hourly measurements comparison.*

**Section 3.4: I think that section could be improved by including some recommendations based on your findings from this study.**

The original section was removed from the manuscript, instead we did implement it as part of section 3.5"Underlying Differences and Future Implications". As suggested by the reviewer we added several recommendations, this includes but not limited to how to use the measurements of the PA-II units, the necessity of Temperature and Relative Humidity measurements, steps for assuring the unit integrity and more.

This information was added to the manuscript:

[revised manuscript text omitted]

**Page 14, line 433: please call it instrument drift, as instrument efficiency is meaningless in this context.**

This sentence was removed from the manuscript.